# Neural correlates of evidence accumulation during value-based decisions revealed via simultaneous EEG-fMRI

M. Andrea Pisauro[1], Elsa Fouragnan[1,2], Chris Retzler[1,3] & Marios G. Philiastides[1]

Current computational accounts posit that, in simple binary choices, humans accumulate evidence in favour of the different alternatives before committing to a decision. Neural correlates of this accumulating activity have been found during perceptual decisions in parietal and prefrontal cortex; however the source of such activity in value-based choices remains unknown. Here we use simultaneous EEG–fMRI and computational modelling to identify EEG signals reflecting an accumulation process and demonstrate that the within- and across-trial variability in these signals explains fMRI responses in posterior-medial frontal cortex. Consistent with its role in integrating the evidence prior to reaching a decision, this region also exhibits task-dependent coupling with the ventromedial prefrontal cortex and the striatum, brain areas known to encode the subjective value of the decision alternatives. These results further endorse the proposition of an evidence accumulation process during value-based decisions in humans and implicate the posterior-medial frontal cortex in this process.

[1] Institute of Neuroscience and Psychology, University of Glasgow, Glasgow, UK. [2] Department of Experimental Psychology, University of Oxford, Oxford, UK. [3] Department of Behavioural & Social Sciences, University of Huddersfield, Huddersfield, UK. Correspondence and requests for materials should be addressed to M.G.P. (email: marios.philiastides@glasgow.ac.uk).

Many decisions in life are based on personal preferences. For example when ordering a dessert at a restaurant, one needs to decide whether one prefers chocolate cake or ice cream. What is the mechanism and source of this deliberation process and how does it differ from decisions based primarily on perceptual evidence (that is, choosing the larger of the items)?

Perceptual decisions are typically characterized both computationally and experimentally in terms of an integrative mechanism whereby information supporting different decision alternatives accumulates over time until an internal decision boundary is reached[1–4]. Evidence supporting this mechanism comes from recent electroencephalography (EEG) studies in humans, which report that electrical activity measured on the scalp builds up gradually over time during perceptual decisions[5–9], and from functional magnetic resonance imaging (fMRI) studies, which propose that this activity is generated in sensorimotor and higher-level prefrontal areas[10–15]. Recent modelling studies also implicate this integrative mechanism in value-based choices[16,17] proposing that evidence accumulation (EA) could represent a domain-general decision processing stage[1,18–20]. However, direct evidence of such accumulating activity in the human brain during value-based decision making is still lacking.

A recent EEG study provided the first evidence of a gradual build-up of activity (in the gamma frequency band) consistent with an accumulation process in a value-based decision making task[21]. Due to the diffuse and macroscopic nature of scalp potentials, however, the source of such activity remains unknown[22]. We hypothesize that if the relevant accumulator regions in the brain exist, then an electrophysiologically derived measure of the process of EA should covary on a trial-by-trial basis with activity in these regions.

To test this hypothesis, we coupled high temporal resolution, single-trial, EEG with simultaneously acquired fMRI[23,24] and computational modelling to (1) uncover the process of EA in the broadband EEG signal and (2) confirm its presence by localizing its source with fMRI during a value-based decision-making task. In doing so, we first identified centroparietal EEG signals exhibiting accumulation-like dynamics and we subsequently demonstrated that within- and across-trial variability in these signals explained fMRI responses in posterior-medial frontal cortex (pMFC). Moreover, we provided evidence that the pMFC exhibits task-dependent coupling with the ventromedial prefrontal cortex (vmPFC) and the striatum, further implicating this region in accumulating information for value-based decisions.

## Results

**Model-based EEG reveals process of EA.** We asked twenty-one hungry participants to choose between pairs of previously rated snack items and indicate their choice with a button press (Fig. 1a). The value difference (VD) in the ratings of the presented items controlled the overall difficulty of the decision. On average, accuracy (choosing the item with the highest rating) increased and reaction times (RT) decreased as the VD of the items increased (Fig. 1b; accuracy: t-test, $t(20) = 15.7$, $P < 0.001$; RT: t-test, $t(20) = -3.95$, $P < 0.001$).

We fitted a dynamical sequential sampling model (SSM) (that assumes a leaky accumulation-to-bound process)[25] to the behavioural data of each individual participant (accuracy: $r = 0.96$, $t(83) = 31.0$, $P < 0.001$; RT: $r = 0.91$, $t(83) = 19.9$, $P < 0.001$; Supplementary Fig. 1a–c) to generate predictions for the average individual temporal profiles of the underlying EA activity (Fig. 1c, Supplementary Fig. 1d). We used these predictions to identify clusters of EEG activity, which exhibited accumulation-

like dynamics leading up to the decision time (Fig. 1c; Methods). We found one such cluster encompassing a set of midline centroparietal electrodes (Fig. 1c, inset) the activity of which correlated significantly with the model predictions ($r = 0.68 ± 0.15$, $t(188) = 30.9$, $P < 0.001$, average correlation across all subjects and electrodes in the cluster; $r = 0.90 ± 0.07$, $t(20) = 61.9$, $P < 0.001$, average correlation across subject-specific electrodes exhibiting the highest correlation with their corresponding model-predicted accumulation profile—henceforth, 'best' electrodes; Supplementary Fig. 1e).

Additional response-locked analyses revealed that the accumulation activity of these electrodes was, on average, modulated by response times (slower accumulation for longer RTs) and task difficulty (slower accumulation in more difficult trials) while approaching a common boundary at time of decision (Fig. 2a,b). These modulations were more pronounced along the RT dimension compared to the VD dimension. We posit this is due to the lack of an objectively consistent trial-wise difficulty level as a consequence of the subjective (and likely noisy) nature of the item preference ratings themselves. Finally, the quality of the EA predicted individual behavioural performance, whereby participants with higher average accumulation rates exhibited better overall performance (higher accuracy) on the task (Fig. 2c).

To rule out that the observed accumulating activity was driven by the perceptual processing of the stimuli themselves, we ran a separate EEG experiment in which participants passively viewed pairs of the same stimuli used in our main value-based task (that is, a task in which a decision was no longer required). We found that the same sensors capturing a gradual build-up of activity in the value task, no longer exhibited accumulation-like dynamics (Supplementary Fig. 2a,b). When, instead, we asked participants to perform perceptual judgments (that is, which item is larger) we found accumulation activity comparable to that observed during value-based choices (Supplementary Fig. 2c,d), confirming that centroparietal scalp activity underlies decision-related EA.

**EEG-informed fMRI reveals source of EA.** Having established a concrete link between EEG activity and EA, we used the signal in the subject-specific best electrodes to provide an electrophysiologically derived trial-by-trial representation of the temporal dynamics of the process of EA. We exploited this endogenous variability to build EEG-informed fMRI predictors to identify whether and where the accumulation process is encoded in the brain. Specifically, for each trial we used the raw EEG time series in the time interval over which the process of EA unfolded (see Methods) to parametrically modulate our fMRI regressor amplitudes (Fig. 3a). We note that trials with lower accumulation rates that require prolonged integration times to reach the decision boundary will have larger areas under the accumulation process[26]. This is consistent with a negative relationship between the slope of EA and the hemodynamic response in the relevant areas[14,27] (see ref. 28 for a good discussion on the different hypotheses about this relationship). Correspondingly, brain region(s) reflecting EA should appear more hemodynamically active in trials with longer compared to shorter integration times (Fig. 3b).

Crucially, the slopes of this accumulating activity were not strongly predictive of individual RTs ($r = -0.13$, $P < 0.005$), due to the high degree of inter-trial variability in the decision and motor planning stages (that is, due to the stochastic nature of these processes). Sequential sampling models of speeded decision making have shown that as a consequence of this variability the shortest RTs end up being approximately the same across all rates of EA, with the longest RTs being somewhat more predictive of the

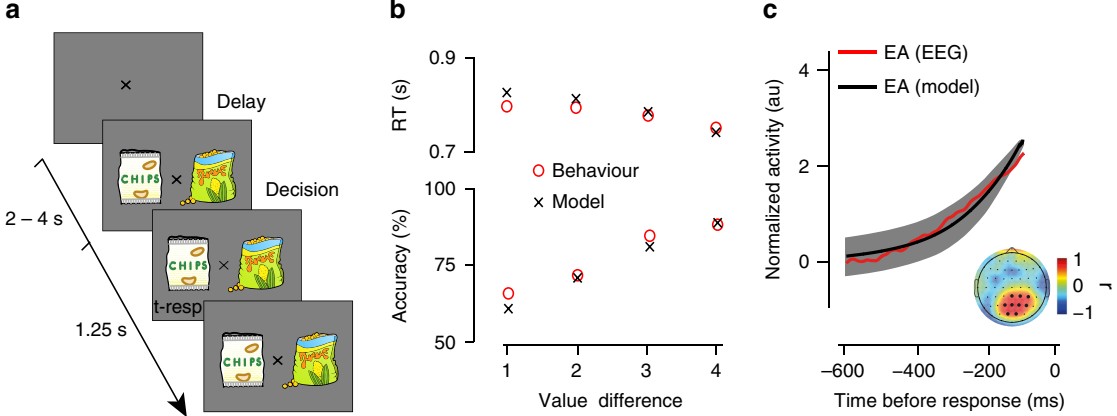

**Figure 1 | Task design, behavioural and modelling results and EEG. (a)** Schematic representation of the experimental paradigm. After a variable delay (2–4 s), two stimuli (snack items) were presented on the screen for 1.25 s and participants had to indicate their preferred item by pressing a button. The central fixation dimmed briefly when a response was registered. Snack stimuli shown here are for illustration purposes only. Participants viewed real branded items during the experiments. **(b)** Behavioural performance (red circles) and modelling results (black crosses). Participants' average (N = 21) reaction time (RT) and accuracy (top and bottom respectively) improved as the value difference (VD) between the alternatives increased. A sequential sampling model that assumes a noisy moment-by-moment accumulation of the VD signal fit the behavioural data well. **(c)** Average (N = 21) model-predicted evidence accumulation (EA) (black) and EEG activity (red) in the time window leading up to the response (on average, 600–100 ms prior to the response), arising from a centroparietal electrode cluster (darker circles in the inset) that exhibited significant correlation between the two signals (see Methods). Shaded error bars represent standard error across participants.

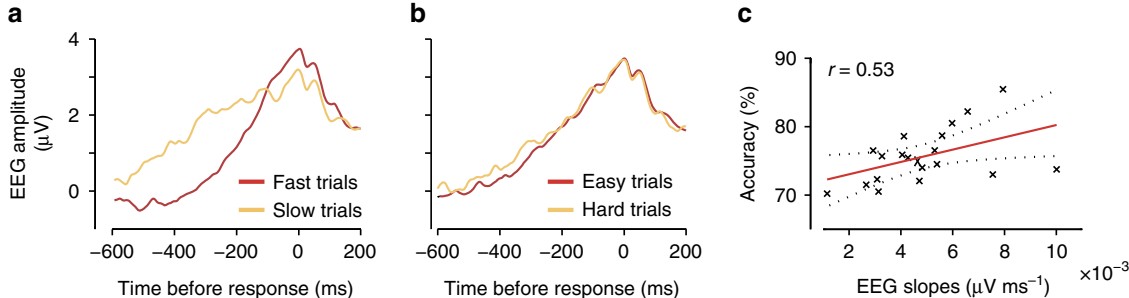

**Figure 2 | Average EEG accumulation dynamics as a function of response time, task difficulty and performance. (a)** Participants' average (N = 21) response-locked EEG activity from subject-specific best electrodes in the centroparietal cluster in fast and slow trials (defined in terms of the median RT). **(b)** Participants' average (N = 21) response-locked EEG activity from subject-specific best electrodes in the centroparietal cluster in easy and hard trials (defined in terms of the median item value difference). **(c)** Across subject correlation between the linear slopes of the average response-locked EEG activity over all trials and individual behavioural performance on the task (robust correlation obtained using Wilcox percentage bend correlation, dotted lines: 95% bootstrapped correlations confidence interval).

accumulation rate (that is, increase as accumulation rates decrease)[29,30]. Correlating separately short and long RT trial groups in our task (by a medial split on RTs) with the individual trial slopes of our EEG activity led to the same observation ($r = -0.06$, $P = 0.95$, for short RTs; $r = -0.21$, $P < 0.001$; for long RTs) suggesting that individual RTs cannot be used to reliably index the rate of EA for individual trials (Supplementary Fig. 3a). Finally, we also showed that the slopes of our accumulating EEG activity were independent from trial-by-trial fluctuations in attention, as indexed by pre-stimulus EEG power in the α-band[31] and further confirmed by the absence of a serial autocorrelation in slopes across neighbouring trials (Supplementary Fig. 3b). Nonetheless, to account for these potentially confounding processes we included separate fMRI predictors for early visual processing, choice/task difficulty and RTs (Fig. 3a; Methods).

Using this fMRI analysis design, we found a cluster in pMFC that was uniquely covarying with the variability—both within and across trials—in our EEG-derived predictor ($Z > 2.57$, cluster corrected using a resampling procedure—see Methods; Fig. 3c; Supplementary Table 1), implicating this region in the process of

EA in value-based choices. Critically, in a supplementary analysis, the EEG variability in short RT trials (those in which the rate of EA is entirely decoupled from the RTs as indicated above) continued to be predictive of activity in the pMFC (Supplementary Fig. 3c,d). Similarly, activity in the pMFC could not be explained better by the inclusion of a separate predictor encoding the decision boundary itself (Supplementary Fig. 3e). These control analyses further confirm that activity in this region cannot be purely explained by the impending motor response or by setting the decision boundary alone but rather by considering the trial-by-trial decision dynamics as a whole.

Finally, our pMFC cluster was not observed in the remaining fMRI regressors, indicating that our electrophysiologically derived predictor offered additional explanatory power than what was already conferred by our stimulus and behaviourally derived regressors (paired t-tests, all $P < 0.05$). Instead, the latter regressors exposed other areas associated with stimulus/value processing, task difficulty and motor execution, consistent with previous reports on value-based decision making[32] (Supplementary Table 1).

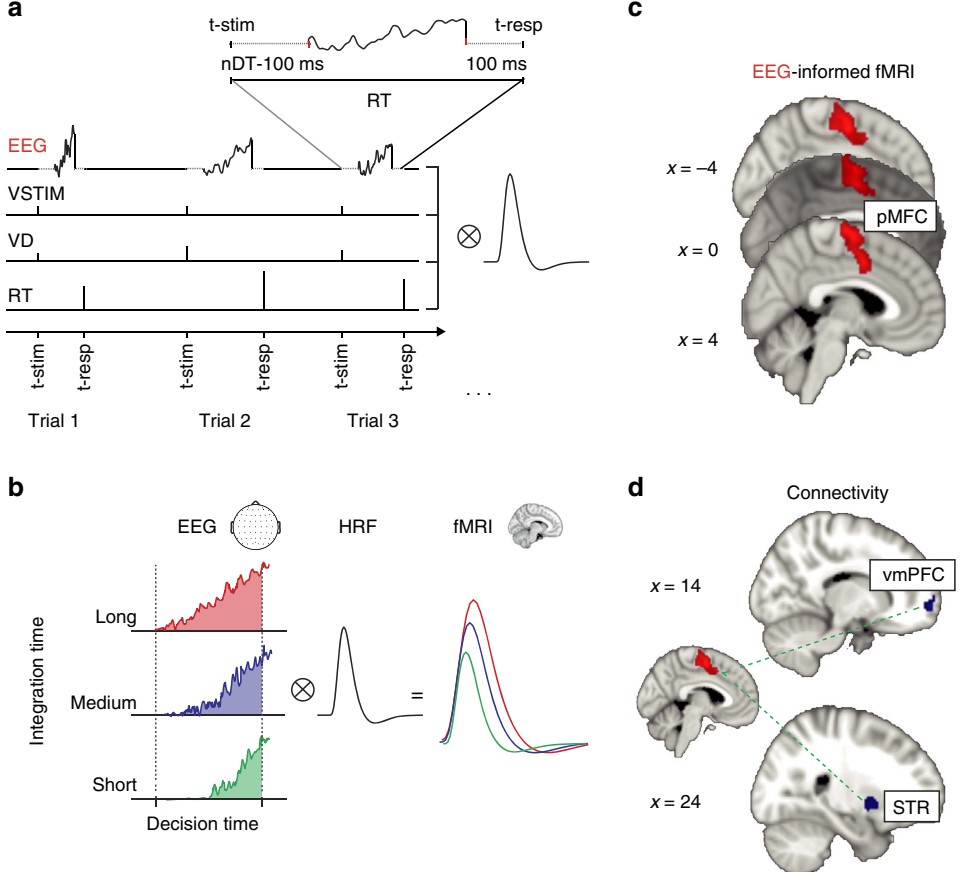

**Figure 3 | EEG-informed fMRI and connectivity analyses in the value-based task.** (**a**) The fMRI GLM model included an EEG-informed regressor capturing the electrophysiological trial-by-trial dynamics of the process of evidence accumulation (EA) in each participant. Three actual single-trial EEG traces are shown. The traces cover the entire trial excluding the time intervals accounting for stimulus processing and motor execution (see inset and Methods for details). To absorb the variance associated with other task-related processes we included three additional regressors: VSTIM—an unmodulated stick function regressor at the onset of the stimuli, VD—a stick function regressor at the onset of stimuli that was parametrically modulated by the value difference between the decision alternatives and RT—a stick function regressor aligned at the time of response and modulated by RT. (**b**) Hypothetical EA traces in response-locked EEG activity ramping up with different accumulation rates. Convolving these traces with a hemodynamic response function (HRF) leads to higher predicted fMRI activity for longer compared to shorter integration times (that is, the predicted fMRI response scales with the area under each EA trace). (**c**) The EEG-informed fMRI predictor of the process of EA revealed an activation in pMFC. (**d**) PPI analysis using the pMFC cluster identified in **c** as a seed revealed an inverse coupling with a region of the vmPFC and the STR. All activations represent mixed-effects and are rendered on the standard MNI brain at $|Z| > 2.57$, cluster-corrected using a resampling procedure (minimum cluster size, 76 voxels).

**pMFC functional coupling and task-general EA**. We reasoned that if the pMFC is indeed related to a process of value-based EA, it should additionally show a task-dependent connectivity pattern with regions of the human valuation system[27] that are known to encode the relevant evidence used in the decision (that is, the absolute difference in value between the two decision alternatives)[27,33,34]. We therefore hypothesize that the coupling with pMFC should be negative, as high VDs decrease integration times and correspondingly the overall integrated activity[35] (that is, area under accumulation curve; Fig. 3b). To this end we ran a psychophysiological interaction analysis with the pMFC as seed. This analysis revealed a significant negative coupling (by VD in the decision period) between the pMFC and two clusters in the vmPFC and the striatum (STR, Fig. 3d), both of which have repeatedly been implicated in valuation[34,36–38] and were indeed modulated by VD in our task (Supplementary Table 1). Intriguingly, this finding is corroborated by recent resting-state connectivity reports showing negative BOLD correlations between regions of the pMFC and ventromedial prefrontal and orbitofrontal cortices[39].

To test whether the pMFC accumulates evidence independent of the task at hand, we ran a separate EEG–fMRI experiment using a probabilistic reward-based decision-making task[23] (with the same participants and setup, Supplementary Fig. 4a). This experiment produced an independent dataset to validate the presence of an accumulation-like activity in the same (best) EEG electrodes that exhibited such activity in our original preference-based choice task. Using this new dataset we found a comparable build-up of activity in the EEG (Supplementary Fig. 4b) that was also predictive of fMRI responses in the pMFC (Supplementary Fig. 4c). These findings suggest that a process of EA drives a range of value- and reward-based decisions and that pMFC might be a common module for driving this process.

**Discussion**
We combined computational modelling and simultaneous recordings of EEG and fMRI to identify a cortical area in pMFC reflecting EA during value-based decisions. We further showed that during decision formation this area was functionally coupled

with brain regions of the human valuation system while it continued to exhibit accumulation-like dynamics during an independent reward-based decision-making task. Taken together, these results support the hypothesis of an EA process in human pMFC underlying a range of value-based decisions.

Recent modelling and stand-alone fMRI studies have made significant progress in establishing a link between value-based decisions and an accumulation-to-bound mechanism[16,26,27,40–42]. The majority of these studies, however, used indirect stimulus- or model-derived correlates of EA that do not necessarily reflect endogenous trial-by-trial variability in information processing, which has been shown to offer additional explanatory power in analysing functional brain imaging data and exposing latent brain states[23,43,44].

A fundamental feature of human decision making is that our responses are variable in the choices we make and in the time it takes for us to make them, even when we are faced with identical decisions on repeated occasions. Computational models of decision making often consider this variability when estimating internal components of processing (for example, accumulation rates). However, most models only produce estimates of the mean and variance of the relevant decision variables across many trials with only a few recent studies attempting to derive single-trial parameter estimates[45–47] of such variables.

The novelty of our work stems from the fact that we captured this trial-by-trial variability by capitalizing instead on an electrophysiologically derived (that is, endogenous) signal of EA and by exploiting the moment-by-moment changes in this signal as the decision process unfolds (that is, we exploited variability both within and across trials). While we used a computational model to select and constrain which features (electrodes, time window) of the EEG data to consider, our approach differs from conventional model-based fMRI in that we do not make any a priori assumptions about which characteristic of the EEG response is relevant (for example, the slope or the boundary of the accumulation) but rather consider the full temporal dynamics of the decision process to capture all relevant variability which could potentially explain the fMRI signal. In other words, our approach allowed us to effectively consider both the drift (that is, average accumulation rate) and the diffusion (that is, the noise) component of the decision process to identify the spatial locus of EA in value-based decisions.

Our EEG measure of EA arose from a cluster of centroparietal electrodes which have also been found to encode decision signals in a wide variety of perceptual tasks and sensory modalities[5,6,8,48,49]. Correspondingly, in a supplementary EEG experiment in which participants performed perceptual judgments (that is, which item is larger) using pairs of the same stimuli used in our original value-based task we found analogous accumulation dynamics in the same EEG electrode cluster. Such supramodal signals are understood to be a signature of the formation of perceptual decisions and are thought to be closely related to the classic P300 (refs 6,50). While we cannot rule out the possibility that additional sources contribute to the generation of this EEG signature[51], our results suggest that the involvement of pMFC in decision formation might span both perceptual and value-based decisions.

The cluster in pMFC we identified here lies on the medial surface of the juxtapositional lobule cortex[52] and extends ventrally to the cingulate cortex bilaterally[53,54]. These subdivisions cover the caudal part of Brodmann's area 6 and 24 respectively[54,55] and are commonly referred to as supplementary motor area (SMA)[56] and posterior mid-cingulate cortex. Both of these areas are traditionally thought to be involved in motor control and preparation of voluntary actions but their precise function remains elusive[57–60].

More recently, these regions were linked to a wide range of other functional roles[58,61] ranging from learning of stimulus–response associations[62], reward prediction error processing[63,64] time perception even in the absence of overt motor responses[65] and value comparison[66]. Correspondingly, bidirectional connections between the SMA and the posterior mid-cingulate cortex were also reported[67] suggesting these areas might act as a single functional unit in a wide variety of tasks[68–70]. Intriguingly, a region partially overlapping with our cluster in the pMFC (including adjacent structures such as the pre-SMA) has also been implicated in adjusting decision boundaries during perceptual EA[71–73].

Therefore, one potential alternative interpretation of our findings is that the activity in this area is related instead to trial-by-trial boundary adjustments. To investigate this further, we computed EEG-derived single-trial boundaries (that is, EEG amplitude differences between the onset and offset of accumulation) and we included these estimates as an additional predictor in a separate fMRI analysis. We found that the activation in the pMFC remained attached to our original EEG regressor capturing the full temporal dynamics of the decision process rather than being absorbed by the new decision boundary regressor. We view these results as additional evidence that the region of the pMFC we reported here cannot be explained purely based on boundary adjustments but rather by considering the decision dynamics as a whole. As such this region appears somewhat different both in location and functional role from those reported in the perceptual decision-making literature.

Taken together, our findings raise the interesting possibility that, at least under conditions of increased urgency to commit to a choice, decisions are encoded in the same sensorimotor areas guiding the actions that implement the choice (that is, embodiment of the decision, here in pMFC). Many electrophysiological and neuroimaging studies of perceptual decision making in humans and monkeys have found choice-predictive activity consistent with an accumulation of sensory evidence in motor[74] and sensorimotor areas[12,75,76], consistent with this interpretation. Our results suggest that a similar mechanism might also operate during value-based decisions whereby activity in pMFC might reflect an increased tendency to select the appropriate motor response. In turn, this tendency could integrate the evidence about the value of the different options encoded in the human valuation system, which appears to be functionally coupled with pMFC before decisions.

In conclusion, our results provide critical new insights regarding the role of pMFC in value-based decision making, complementing previous reports that have implicated this region in perceptual decisions[14,71–73]. Our general research approach of combining computational modelling with simultaneous EEG/fMRI recordings opens up new avenues for a more targeted investigation of the neural systems underlying value-based decision making in humans. Our findings also have the potential to further improve our understanding of how everyday decisions can sometimes go astray and how such maladaptive behaviours can affect reward learning and strategic planning.

## Methods

**Participants.** Twenty-four subjects participated in the experiment. Three were removed for excessive head movements inside the scanner. The remaining subjects (8 males, 13 females), aged between 18 and 31 years (mean = 22 years, s.d. ± 2.5), were included in all subsequent analyses. They were all right handed, had normal or corrected-to-normal vision and reported no history of psychiatric, neurological or major medical problems, and were free of psychoactive medications at the time of the study. Written informed consent was obtained in accordance with the School of Psychology Ethics Committee at the University of Nottingham.

**Stimuli and behavioural task.** The behavioural task consisted of two steps: (1) a rating phase (outside of the MR scanner) and (2) a speeded two-choice decision-making task (inside the MR scanner). In the rating phase, we asked participants to provide a subjective value rating for 80 different snack items. Before providing the ratings, subjects briefly saw all of the items for an effective use of the rating scale. Participants indicated how much they liked to eat each snack using an on-screen Likert scale ranging from $-5$ (really dislike) to 5 (really like) with unitary increments.

The main decision-making task followed shortly thereafter (Fig. 1a). Trials started with the presentation of a central fixation cross (subtending $0.6° \times 0.6°$ of visual angle) that served as an inter-stimulus interval (ISI: in the range of 2–4 s). Subjects were instructed to focus on the central fixation. Following the ISI two food items were simultaneously displayed to the left and to the right of the fixation cross (subtending $\sim 3° \times 3°$ of visual angle) for 1.25 s and participants were asked to respond within this time period and indicate the item they preferred the most. Participants indicated their choice by pressing the left or right button on a fORP MRI compatible response box (Current Design Inc., Philadelphia, PA, USA) using their right index or middle finger, respectively. After making a choice, the fixation cross dimmed briefly (100 ms) to signal successful registration of the response. Trials in which participants failed to respond within 1.25 s of stimulus presentation were followed by a 'lost trial' message and were excluded from further analysis. There was no cost for lost trials and overall these were extremely rare (<1% of all trials). We defined a correct response as a choice in which the subject selected the item with the highest rating. Participants were required to maintain fixation throughout the trial.

We manipulated the difficulty of the task by controlling the VD between the two presented items (based on the original subject-specific ratings). We constructed random pairs of items and constrained the VD to one of four possible levels [1, 2, 3, 4]. We note that across participants the VD of all items pairs was virtually decoupled (zero correlation) from subtle perceptual differences in the stimuli, such as differences in size, luminance or contrast. Each experiment consisted of 400 trials (100 trials per VD level) divided in two blocks of 200 trials each. Trials were presented in a fully interleaved manner. Participants were instructed to refrain from eating in the 3 h leading up to the experiment and were told that one of their item choices during the main task would be randomly selected for them to consume in the lab at the end of the experiment. To test whether VD modulated behaviour we computed subject-specific linear regression coefficients for VD versus Accuracy and VD versus RT and performed separate two-tailed $t$-tests on these coefficients.

The fixation cross and the stimuli were equated for luminance and contrast. A Windows Professional 7, 64 bit-based machine (3 GB RAM) with an nVidia (Santa Clara, CA) graphics card and Presentation software (Neurobehavioral Systems Inc., Albany, CA) controlled the stimulus display. An EPSON EMP-821 projector (refresh rate: 60 Hz, resolution: 1,280 × 1,024 pixels) projected the images onto a screen, which was placed 2.3 m from the subject (projection screen size: $120 \times 90$ cm).

**Sequential sampling modelling.** Following a recent study[21] we modelled evidence accumulation (EA) as a Ornstein–Uhlenbeck process , which represents a special case of the leaky competing accumulator family of models[25]. This process is described by the equation:

$$EA(t+1) = EA(t) + (\lambda EA(t) + k VD)dt + N(0, \sigma), \qquad (1)$$

where VD is the value difference which drives the accumulation (that is, difference in value between the food items), $k$ is a parameter that modulates the input, $\lambda$ is a parameter that denotes the leak strength (or acceleration to threshold) of the process and $N(0, \sigma)$ is a Gaussian noise term with standard deviation $\sigma$. We used $dt = 0.001$ s and assumed that the model makes a decision when $|EA| > 1$ (that is, setting the decision threshold for a correct and error response to $+1$ and $-1$ respectively; Supplementary Fig. 1a). We accounted for early visual encoding of the stimuli and motor preparation by adding a non-decision time (nDT) (a free parameter in the model) to the time taken to reach the threshold. The model was fitted to the individual participants' RT data (Supplementary Fig. 1b) using a maximum likelihood estimation. Specifically, RTs were separated into correct and error trials for each of the four VD levels. RTs from correct and incorrect trials were then combined into a single distribution by mirroring the distribution of incorrect trials at the zero point along the time dimension, so that all the times in this distribution received a negative sign[77]. This RT distribution and participants' choice accuracies were compared to the RT distribution and proportion of correct choices generated by the model. For a given set of parameter estimates, we estimated the log likelihood (LL) of the data using the following formula:

$$LL \sim \sum_{VD=1}^{4} \log\left( KS\left( RT_{data}^{VD}, RT_{model}^{VD} \right) \right) + \sum_{VD=1}^{4} \log\left( \exp\left( -\left( \frac{Accuracy_{data}^{VD} - Accuracy_{model}^{VD}}{0.1} \right)^2 \right) \right), \qquad (2)$$

where $KS(p,q)$ estimates the probability that two distributions are equal according to the Kolmogorov–Smirnov test (computed using MATLAB function *ktest2* which in turn estimates the predicted cumulative probability through the proportions of the predicted RTs which are less than or equal to any observed RT), VD represents

a given difficulty level and accuracies are computed as proportion of correct response for the data and the model for each difficulty level. For each participant separately, we identified the set of model parameters that maximized the LL, by searching over a grid of values: $\lambda = (2.5, 3.0, \ldots, 7, 7.5)$, $k = (0.02, 0.04, \ldots, 0.4)$, $\sigma = (0.003, 0.006, \ldots, 0.024)$ and nDT $= (0.1, 0.15, \ldots, 0.6)$ s. These ranges were defined after an initial exploratory analysis over a wider range of parameter values to ensure selecting the ones that produced choice accuracies spanning those seen in behaviour. For each set of parameters we generated RT distributions and choice accuracies by running 5,000 simulations of the model (that is, by producing decision trajectories using equation (1)) for each difficulty level. To further assess the quality of the fits resulting from the best set of subject-specific parameters (those that maximized the LL function in equation (2)), we computed correlation coefficients between the average accuracy and RT from the data and the model for all participants and VD levels. We also performed a separate parameter recovery analysis[78] to ensure that the parameters of our SSM were estimated reliably (Supplementary Fig. 5, Supplementary Methods).

Next, we applied the best set of subject-specific parameters to generate individual participants' model-predicted EA signals (Supplementary Fig. 1c) by averaging activity of all the simulated trials time-locked to the decision, starting $-1.25$ s before the decision threshold is crossed. If the response time of the model was shorter than 1.25 s, then we padded the beginning of the trial with null values (that is, these values did not contribute to the average across simulated trials). Averages of model-predicted activity were quantitatively tested against average EEG response-triggered traces for each participant individually (see subsection on EEG data analysis below).

Finally, we note that we also tested alternative SSM models, which either included an additional threshold parameter $\theta$ to account for potential variability in the decision boundary across participants (Supplementary Methods) or excluded the leak term $\lambda$ all together. However, these alternative models did not provide a better fit to the data compared to our original model (Supplementary Fig. 5). For the former this is likely due the difficulty of identifying the threshold parameter $\theta$ together with drift rate and noise while for the latter due to the exponential nature of the EA signal (Supplementary Fig. 3e). Critically, however, since we are using the EEG signal itself as a regressor for the fMRI analysis we ensured that potential misspecifications in the model have a lesser influence on the eventual inference.

**EEG data collection.** EEG data was acquired at a 5-kHz sampling rate at the same time as the fMRI data collection, using an MR-compatible EEG amplifier system (BrainAmps MR-Plus, Brain Products, Germany) and the Brain Vision Recorder software (BVR; Version 1.10, Brain Products, Germany). Data were filtered online with a hardware band-pass filter of 0.016–250 Hz. The EEG cap included 64 Ag/AgCl scalp electrodes which were localized according to the international 10–20 system. The reference electrode was positioned between electrode Fpz and Fz while the ground electrode was positioned between electrode Pz and Oz. All electrodes had in-line 10 kΩ surface-mount resistors to ensure subject safety. All leads were bundled together and twisted for their entire length to minimize inductive pick-up and maximize participant's safety. Input impedances were kept below 20 kΩ (including the 10 kΩ surface-mount resistors on each electrode). EEG data acquisition was synchronized with the fMRI data (Syncbox, Brain Products, Germany) and triggers from the MR-scanner were collected separately to remove MR gradient artifacts offline. Scanner trigger pulses were lengthened to 50 µs using an in-house pulse stretcher to facilitate accurate capture by the BVR. Experimental event codes were also synchronized with the EEG data and collected using the BVR software. MR gradient artifacts were minimized by ensuring that electrodes Fp1 and Fp2 were at the isocentre of the MR scanner in the $z$-direction (by aligning these two electrodes with the laser beam used to place the participants inside the bore). We used a 32-channel SENSE head coil which presented an access port at the top of participants head, allowing the EEG cap cables to run along a straight path out of the scanner. This manipulation ensured no wire loops, thus minimizing the risk of RF heating of the EEG cap and associated cables and of inducing EEG artifacts. To additionally minimize induced artifacts, the cabling was isolated from scanner vibrations as much as possible, through the use of a cantilevered beam[79].

**EEG pre-processing.** We performed EEG pre-processing offline using MATLAB (Mathworks, Natick, MA). EEG signals recorded inside an MR scanner are contaminated with gradient and ballistocardiogram (BCG) artifacts due to magnetic induction on the EEG leads. We first removed the gradient artifacts. Specifically, from each functional volume acquisition we subtracted the average artifact template constructed using the 80 volumes centred on the volume-of-interest using in-house MATLAB software. We repeated this process for as many times as there were functional volumes in our data sets. We subsequently applied a 10-ms median filter to remove any residual spike artifacts. Next, we band-pass filtered the data by applying a 0.5-Hz high-pass filter to remove direct current (DC) drifts and a 40 Hz low-pass filter to remove high frequency artifacts not associated with neurophysiological processes of interest. These filters were applied together, non-causally to avoid distortions caused by phase delays.

BCG artifacts share frequency content with the EEG and as such are more challenging to remove. To avoid loss of signal power in the underlying EEG we adopted a conservative approach and removed a small number of BCG

components using principal component analysis in two steps. Firstly, four BCG principal components were extracted from data that were initially low-pass filtered at 4 Hz to extract the signal within the frequency range where BCG artifacts are observed. Secondly, the sensor weightings corresponding to those components were projected onto the broadband (original) data and subtracted out.

**Eye-movement artifact removal.** Before the beginning of the fMRI acquisition, participants performed an eye-movement calibration task during which they were asked to blink repeatedly on the appearance of a fixation cross in the middle of the screen and to make several horizontal and vertical saccades by following a fixation cross moving right to left and up and down on the screen, respectively. The fixation cross subtended $0.6° × 0.6°$ of visual angle while the horizontal and vertical saccades subtended 30° and 22°, respectively. Using principal component analysis we determined linear EEG sensor weightings corresponding to these eye blinks and saccades, which we then projected onto the broadband data from the main task and subtracted out.

**EEG data analysis.** We computed EEG response-triggered traces for all subjects and electrodes by averaging together all trials in the interval ranging from 700 ms before to 200 ms after response time. We excluded noisy trials in which more than ten electrodes had an average trial amplitude above 2 s.d. from the grand mean across electrodes in the time range above (on average, we removed $<5\%$ of the total number of trials). To identify clusters of EEG activity that exhibited accumulation-like dynamics we regressed these EEG traces (for each participant individually) against the subject-specific average model-predicted EA response (equation (1); Supplementary Fig. 1c). For this analysis, we focused on a time window starting 600 ms prior to the response (when build-up of activity started, on average, to unfold) and lasting until 100 ms before the response. We also estimated individual trial EEG slopes (using trial-specific time windows) for a separate EEG-informed fMRI analysis (see 'fMRI analysis' section below for details). We purposely excluded the last 100 ms leading up to the response to avoid potential confounds with activity related to motor execution (due to a sudden increase in corticospinal excitability in this period[80]). We selected electrodes, which survived a one-sample two-sided $t$-test of the population of regression coefficients computed individually above, with a significance level of 0.05 (Bonferroni-corrected by the number of EEG electrodes). We considered clusters comprising of at least three significant neighbouring electrodes. This analysis led to the identification of a cluster of nine midline centroparietal electrodes.

We computed correlations between the activity of each of these electrodes and the EA profile produced by the model by considering the average response-locked EEG activity for each individual subject. We used this approach to identify subject-specific best sensors and computed their average correlation. We also computed the average correlation in the entire centro-parietal electrode cluster by averaging the data/model correlations across all participants and electrodes in the cluster. To build subject-specific fMRI predictors of the process of EA (see subsection on fMRI analyses below) we used, for each participant, the best electrode within this cluster that showed the highest correlation with the model's EA response (Supplementary Fig. 1d,e).

**MRI data collection.** We acquired the fMRI data using a 3T Philips Achieva MRI scanner (Philips, Netherlands). Specifically, we collected functional Echo-Planar-Imaging (EPI) data using an 32-channel SENSE head coil with an anterior–posterior fold over direction (SENSE factor: 2.3; repetition time: 2.5 s; echo time: 40 ms; number of slices: 40; number of voxels: $68 × 68$; in-plane resolution: $3 × 3$ mm; slice thickness: 3 mm; flip angle: 80°). Slices were collected in an interleaved order. Altogether, we collected two separate runs of 317 volumes each, corresponding to the two blocks of trials in the main experimental task. Anatomical images were acquired using a MPRAGE T1-weighted sequence that yielded images with a $1 × 1 × 1$ mm resolution (160 slices; number of voxels: $256 × 256$; repetition time: 8.2 ms; echo time: 3.7 ms). We also acquired a B0 map using a multi-shot gradient echo sequence which was subsequently used to correct for distortions in the EPI data due to B0 inhomogeneities (echo time: 2.3 ms; delta echo time: 5 ms; isotropic resolution: 3 mm; matrix: $68 × 68 × 32$; repetition time: 383 ms; flip angle: 90°).

**fMRI pre-processing.** We discarded the first ten volumes from each fMRI run to ensure a steady-state MR signal, and we used the remaining 307 volumes for the statistical analysis presented in this study. Pre-processing of our data was performed using the FMRIB's Software Library (Functional MRI of the Brain, Oxford, UK) and included: head-related motion correction, slice-timing correction, high-pass filtering ($>100$ s), and spatial smoothing (with a Gaussian kernel of 8 mm full-width at half maximum). To register our EPI image to standard space, we first transformed the EPI images into each individual's high-resolution space with a linear six-parameter rigid body transformation. We then registered the image to standard space (Montreal Neurological Institute, MNI) using FMRIB's Non-linear Image Registration Tool with a resolution warp of 10 mm. Finally, B0 unwarping was applied to correct for signal loss and geometric distortions due to B0 field inhomogeneities in the EPI images.

**fMRI analyses.** We performed whole-brain statistical analyses of functional data using a multilevel approach within the generalized linear model (GLM) framework, as implemented in FSL through the FEAT module:

$$Y = X\beta + \varepsilon = \beta_1 X_1 + \beta_2 X_2 + \beta_3 X_3 + \beta_4 X_4 + \varepsilon, \quad (3)$$

where $Y$ is the times series of a given voxel comprising $T$ time samples and $X$ is a $T × 4$ design matrix (Fig. 3a) with columns representing four different regressors (see below) convolved with a canonical hemodynamic response function (double-$\gamma$ function). $\beta$ is a $4 × 1$ column vector of regression coefficients and $\varepsilon$ a $T × 1$ column vector of residual error terms. We performed a first-level analysis to analyse each participant's individual runs, which were then combined using a second-level analysis (fixed effects). Finally, we used a third-level, mixed-effects model (FLAME 1) to combine data across subjects, treating participants as a random effect. Time-series statistical analysis was carried out using FMRIB's improved linear model with local autocorrelation correction.

Our GLM model included an EEG-informed regressor capturing the trial-by-trial dynamics of the process of EA. Specifically, for each trial we used the raw EEG time-series (from the subject-specific sensor that was most predictive of the model-derived EA profile) to parametrically modulate the regressor amplitudes. We considered the entire trial duration (that is, RT) minus the subject-specific nDT estimated by the model, which accounted for stimulus processing and motor execution. More specifically, we split this nDT in two intervals by fixing the motor preparation to 100 ms prior to the response (when a sudden increase in corticospinal excitability occurs[80]) and setting the average duration of the stimulus encoding to nDT-100 ms (Fig. 3a). To absorb the variance associated with other task-related processes we included three additional regressors: (1) an unmodulated stick function regressor at the onset of the stimuli, (2) a stick function regressor at the onset of stimuli that was parametrically modulated by the VD between the decision alternatives and (3) a stick function regressor aligned at the time of response and modulated by RT (Fig. 3a). As a control analysis we also removed the RT and VD regressors from the GLM design to test if our EEG-informed regressor absorbed additional activations. The only activation we found in the EEG-informed regressor was the one capturing accumulation dynamics as in the main analysis (that is, pMFC) with a marginal improvement in the statistical significance of the area. Regions previously absorbed by the other regressors moved to our constant term regressor (that is, our unmodulated regressor). This finding suggests that it is truly the endogenous electrophysiological variability in our EEG-derived regressor that is driving the observed effects in the pMFC.

To test whether activity in the pMFC was driven instead by boundary adjustments we performed a separate analysis. We estimated individual trial boundaries directly from the EEG traces (that is, EEG amplitude differences between the onset and offset of accumulation; Supplementary Fig. 3e) and we included these estimates as a separate parametric predictor in our fMRI GLM analysis. We found that the activation in the pMFC remained attached to our original EEG regressor representing the full temporal dynamics of the decision process rather than being absorbed by the new boundary regressor. We formalized this observation by showing that our original regressor was a significantly better predictor of the fMRI signal in the pMFC than the boundary regressor ($t(20) = 4.21$, $P < 0.001$; paired $t$-test comparison of the $\beta$ coefficients in pMFC; Supplementary Fig. 3e).

**Resampling procedure for fMRI thresholding.** To correct the fMRI statistical maps for multiple comparisons, we used a resampling procedure that took into account the a priori statistics of the trial-to-trial variability in all of our fully parametric regressors in a way that trades off cluster size and maximum voxel $Z$-score. Specifically, we maintained the overall distributions of the regressor amplitudes while removing the specific trial-to-trial correlations in individual experimental runs. Thus for each resampled iteration and each regressor type, all trials were drawn from the original regressor amplitudes, however, the specific values were mixed across trials and runs. We repeated this procedure 100 times and for each iteration we run a full three-level analysis (session, subject and group). We then used the cluster outputs from the permutated parametric regressors to establish a joint threshold on cluster size and $Z$-score. Specifically, we considered the sizes of all clusters larger than 10 voxels and surviving a $Z$-score of $|2.57|$ (that is, for positive and negative correlations with the permuted parametric regressors) to build a null distribution for the joint threshold described above. Finally, we used this distribution of cluster sizes and found that the largest 5% of cluster sizes exceeded 76 voxels. We therefore used these results to derive a corrected threshold for the statistical maps of our original data. All fMRI clusters described in our analysis survived this corrected threshold (that is, $Z > 2.57$, minimum cluster size of 76 voxels, corrected at $P < 0.05$).

**Extracting time-series data.** We extracted time-series data from subject-specific pMFC clusters of interest for a psycho-physiological interaction (PPI) analysis (see below). We first drew subject-specific masks of the pMFC based on the overlap between the cluster obtained from the group analysis and the relevant (subject-specific) statistical maps in standard space (second-level analysis). For these statistical maps we used a more lenient threshold of $P < 0.05$ uncorrected, and cluster size $> 10$ voxels to accommodate for inter-subjects variability in statistical power and cluster's location. We subsequently back-projected these clusters from

standard space into each individual's EPI (functional) space by applying the inverse transformations as estimated during registration (see fMRI pre-processing section). Finally, we computed average time-series data from all voxels in the back-projected clusters in each subject to serve as a physiological regressor in the PPI analysis.

**PPI analysis.** Using the procedure described above, we extracted time-series data from individual clusters in pMFC, which served as a seed region (that is, the physiological regressor—PHY) for a PPI analysis. This analysis was primarily designed to investigate the potential interaction of the area encoding accumulation of evidence with brain regions known to encode decision values. In other words, the increase in correlation between pMFC and these regions should be specific for the task in which this coupling is relevant; that is, it should be greater during the time window leading up to response time in which the accumulation of evidence unfolds and scale with the evidence for the decision. Therefore, we constructed our psychological (PSY) task regressor as a parametric regressor boxcar regressor with a step function in the interval between stimulus onset and—response time whose amplitude was modulated by the difference in value between the alternatives (zero otherwise). Correction for multiple comparisons was performed on the whole brain using the outcome of the resampling procedure as described earlier in the Resampling procedure for fMRI thresholding subsection.

**Data availability.** The data that support the findings of this study are available from the corresponding author upon request. The code to generate the results and the figures of this study are available from the corresponding author upon request.

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

## Acknowledgements

This work was supported by the Biotechnology and Biological Sciences Research Council (BBSRC; grant BB/J015393/2 to M.G.P.) and the Economic and Social Research Council (ESRC; grant ES/L012995/1 to M.G.P.). We thank Karen Mullinger for assistance with data collection and Philippe Schyns, Lawrence Barsalou, Joachim Gross and Robin Ince for useful comments on earlier versions of the manuscript.

## Author contributions

E.F., C.R. and M.G.P. designed the experiments. E.F. and C.R. collected the data. M.A.P., E.F. and M.G.P. analysed the data and wrote the paper. All authors discussed the results and implications and commented on the manuscript at all stages.

## Additional information

**Competing interests:** The authors declare no competing financial interests.

