## [Peer Review File · Nature Communications]

Reviewers' comments:

Reviewer #1 (Remarks to the Author):

This is a very exciting paper using simultaneous EEG and fMRI to study evidence accumulation in value-based decision making. The combination of these two techniques helps to overcome each's limitation: poor spatial resolution with EEG, poor temporal resolution with fMRI. The authors utilize an established behavioral paradigm involving choices between foods, though they also appear to replicate the effects with a second learning task. How the brain accumulates and compares evidence in such decisions is a big question and many papers have tackled it using one technique or the other, relying on assumptions about the activity underlying the BOLD signal (with fMRI studies) or conjecturing about the sources of the activity (with EEG studies). In this paper the authors claim to have found a signal, originating in the posterior medial frontal cortex, that shows the expected ramping up of activity in the EEG data. These are certainly exciting findings. What remains unclear though is whether this activity truly reflects value-based evidence accumulation, or some other signal. This, and some other statistical questions, need to be cleared up before I can fully support publication.

(1) The authors fit a sequential sampling model (Ornstein-Uhlenbeck process) to the participants' behavioral data, then compare the model output with EEG activity. However, it appears that the authors, in identifying EEG activity that correlates with the model output, did so using only one EA profile per participant. I don't understand why the authors did it this way, since different trials had different VD and so should have exhibited different average EA curves. Basically, the authors need to do more to convince us that the EEG activity they've identified truly represents evidence accumulation. Other papers in this literature have looked at things like modulation with VD, accumulation to a fixed bound, etc. These are basic tests that should be performed.

On a related point, the reported correlation here is likely inflated due to the fact that the "best" electrode is chosen for each participant. While this does make sense for then correlating it with the fMRI data, it does not give an accurate representation of the true correlation. The authors could perhaps use their other experiment to estimate the correlation out of sample.

On another related point, it is unclear throughout the manuscript how these reported correlations (and their significance) were calculated. There were obviously repeated observations for each participant. So are these correlations run at the individual level and then averaged? Or was some sort of mixed-effects analysis performed?

(2) Why is it "crucial" that the slopes of the accumulating activity are not predictive of RTs (as the authors write near the bottom of p.4)?

The authors seem to be making a distinction between "drift" and "diffusion" and arguing that the drift is being reflected by the EEG activity, but not the diffusion. When they say that the shortest RTs are flat with respect to "rates of evidence accumulation" they mean

that RTs are flat with respect to mean drift rate. Of course RTs are NOT flat with drift + diffusion, its just that we don't normally measure the diffusion part. In any case, from looking at Supplementary Figure 2, it appears that RTs are not flat with respect to the EEG activity. Instead they appear to peak when the EEG slope = 0. The authors should try a correlation between absolute EEG slope and RT.

(3) Given the comments above and the overlap of the pmFC region with the SMA, I have to wonder if the activity that the authors are seeing is instead perhaps related to boundary collapsing or an urgency signal that increases over time, but does not otherwise strongly relate to VD or RT. Given the prior literature linking the SMA and boundary setting, this seems like a rather plausible alternative explanation. I wonder if the authors could address this?

Minor comments

(4) It should be mentioned that there are alternative views about how evidence accumulation should be represented in BOLD activity. Hare et al. 2011 (PNAS) argue for the opposite and find a region in dmpfc that this paper also appears to find. This warrants at least some discussion, if not more analyses. Mulder, van Maanen, and Forstmann (2014) provide an interesting discussion of this issue.

(5) What is the rationale for the relative weighting of choice and RT data in the LL function? Were there 5000 simulations per VD, or overall?

(6) Analyses using RT as the dependent variable should use $\log(\text{RT})$ instead, since RTs are typically log-normal.

(7) Was there any cost of a "lost trial" to the participants?

(8) There are a lot of technical details when relating the EEG data to the fMRI data. This is obviously a critical contribution of the paper, so it would be helpful to point what here has been established already and what methods are novel to this paper.

Reviewer #2 (Remarks to the Author):

I have previously reviewed this manuscript at a different journal and, while many of the original reviewer comments have been addressed, some important ones have not and so they are repeated below here.

In this paper Philiastides et al seek to shed light on the neural origins of value-based decision making through the combination of computational modelling and the fusion of simultaneously recorded EEG and fMRI data. Participants completed a two-alternative relative value judgment task. The authors fit a leaky accumulator model to the behavioural data and then identified EEG electrodes whose trial-averaged response-aligned activity

exhibited a significant correlation with the model-predicted evidence accumulation time-courses. Single-trial activity from these electrodes was then included as a regressor in the fMRI analysis and this approach highlighted the posterior medial frontal cortex. The authors are also able to demonstrate that this same region exhibits negative coupling with ventromedial prefrontal and orbitofrontal cortices- two regions previously implicated in representing value. The authors conclude that the pMFC likely plays a role in motor control and thus represents emerging decisions in a generic fashion, not limited to value-based decisions.

This is an interesting and very nicely written manuscript that brings a new and potentially valuable methodology to bear on an important question. However, I am still not convinced that the strong claims made by the authors are warranted based on these data.

Major Comments:

The significance of the conclusion offered by the present study rests on the idea that there might be a general-purpose accumulator that, just as it accumulates perceptual information for perceptual decisions, accumulates value information for value-based decisions. But the alternative choice items are presented visually, and subjects have to identify the items in order to know their value and compare them. So how can it be determined that any observed buildup dynamics correspond to the accumulation of the value information and not the perceptual information that is required for the more primary task of figuring out what one is looking at? With this in mind it is actually very surprising that only a single brain region is found to correlate with the EEG accumulator activity – where are the perceptual decision regions highlighted in previous work? A recent study by Polania et al (2014) did succeed in dissociating EEG markers of perceptual vs value-based decision making yet these are not considered for analysis here.

The title of this paper and the abstract both strongly emphasise the role of pMFC in value-based decision making yet the authors have no evidence that the pMFC's role is specific to valuation and in fact conclude in the Discussion that it most likely plays a generic role in decision making. I think this is a highly plausible interpretation but, without inclusion of a perceptual decision task and a condition in which participants used an alternative effectorto indicate their decisions, it is something of a speculation. Moreover, this interpretation of pMFC as a generic motor control region calls into question the emphasis in both the title and abstract that the pMFC is the site of value-based decision making which implies that the study has identified a region that is specifically involved in accumulating value. The authors should also consider in more detail the previous extensive fMRI literature on decision making and whether or not the pMFC has been highlighted previously (e.g. Liu & Pleskac J Neurophysiol 2011).

The authors highlight that 'the slopes of this accumulating activity were not strongly predictive of RTs due to the high degree of inter-trial variability in decision processing.' The authors should clarify what they mean by 'decision processing' but a graver concern is that

they go on to include RT as an additional regressor of no interest in the fMRI analysis. Even if the influence of accumulator build-up rate on RT is weak for early trials, how is the cross-trial variation in build-up rate at all meaningful if it can have no impact on behaviour? The inclusion of evidence strength as a regressor-of-no-interest is equally problematic since an accumulator process should also covary with this variable. The analysis is constructed in a way that would systematically tend to remove true evidence accumulation processes from the picture. This might partly account for the rather surprising fact that only one brain region appears to be associated with decision formation on this task.

I also have a concern about the slope measurements. A window starting 600ms before response execution was used which means that for any RTs < 600ms the trial will incorporate activity that precedes stimulus onset. Surely this is problematic and would also lead to meaningless slope measurements on certain trials? Here the paper would benefit from some 'sanity check' analyses such as demonstrating that the build-up rate of the EEG activity is faster for easy vs difficult trials.

Reviewer #3 (Remarks to the Author):

The authors report a study in which BOLD fMRI is predicted using EEG signatures of evidence accumulation. They find that BOLD responses in pMFC reflect a rising (accumulating) signal in a cluster of centroparietally located electrodes. These electrodes were identified earlier by studying the correlation between the hypothesized evidence accumulation signal and the EEG signal. The authors conclude that the results suggest that activity in pMFC reflects a decision formation, contrary to earlier reports that these areas (or areas in close proximity) reflect a decision bound (e.g., Ivanoff et al., 2008; Forstmann et al., 2008, Van Maanen et al., 2011; Van Veen et al., 2008; Winkel et al., 2012). If this conclusion is valid, this finding is exciting and might encourage all of us to think about decision making processes in a lot less static way, as it questions the often assumed dichotomy between "evidence accumulation areas" and "decision bound areas".

From a methodological view point there are many aspects of this paper that I like, but I also have a number of methodological concerns. First of all, I think the independent replication is a strong asset of this study, as it shows the robustness of the results, which is often problematic in (model-based) fMRI studies (e.g., Button et al., 2013). Secondly, I appreciate that the authors chose to include the EEG signal itself as a regressor for the fMRI analyses, rather than a model-based estimate of the evidence accumulation process. This way, a potential misspecification of the evidence accumulation model has less influence on the eventual inference.

My main concerns all relate to the computational modeling part of the paper, and can be divided in two broad categories:

1. The choice of the leaky accumulator model.

An important aspect in this study is the specification of the evidence accumulation model, as

it directly influences the choice of electrode for the fMRI analyses. The authors opted to fit a leaky accumulator model. The inclusion of the leak term is however not supported. Wouldn't a model without leak be a better (more parsimonious) choice? Figure 1c suggests that the normalized EEG activity ramps in a more or less linear fashion. This could be due to averaging across subjects of course, but it could also reflect that the evidence fully integrates (ie., no leak). Another more practical benefit of a model without leak term is that estimating parameters is less error prone (more on this under point 2).

A second choice in the model that I think deserves more scrutiny is the choice to constrain the parameter scaling by fixing the thresholds at 1 and -1. This constraint embodies the strong assumption that individual variability in choice behavior is fully explained by the evidence accumulation process, and not by variability in threshold setting. This is important given the alternative hypothesis that pMFC reflects a decision bound (as eluded to above). If the evidence accumulation parameters (λ and k) absorb variability in choice behavior that is due to individual differences in threshold, then a model that allows these differences might explain the data better, and might alter the interpretation of these findings.

2. The parameter estimation of the accumulator model.

The authors' choice of model requires a simulation of the predicted RT distributions during the parameter optimization procedure, because there exists no closed-form likelihood function of this particular model. Such a simulation may lead to misspecification of parameters, because of noise in the predicted RT distributions. A recent study in my lab (currently under review, but I can send a draft of the manuscript if desired) shows that in particular the leaky competitive accumulator model is prone to this, partly because of strong tradeoffs between the estimates of k and λ (and inhibition, but that is not included in the current model). The correlation of 0.60 between the λ and k parameter presented in Supplementary Table 2 hints that this may be a problem here as well. It could very well be that this is a minor issue in this study, because of the nice constraint on the k parameter (the estimated trial difficulty using individual value estimates of all presented items).

Nevertheless, it would be good to see a parameter recovery analysis.

In addition, I am not convinced that the statistic that is optimized yields the best possible set of parameters. Firstly, I think that the KS statistic does not represent the likelihood of the data under the model, but rather the probability that two distributions are identical. While these two concepts are related, I think it would be good to be precise about what is being optimized.

Secondly, the weighting of the different components of the KS statistic seems a bit arbitrary. Effectively, the influence of the RT distribution of incorrect responses on the summed log KS value is assumed as strong as the RT distribution of correct responses, even though typically there are fewer incorrect responses, and therefore the estimate of this distribution is poorer. Similarly, the estimation of accuracy (one data point per participant/condition) is assumed as important as the estimation of a full RT distribution. Perhaps the weight of the accuracy is scaled by the division of the difference in error rates between data and model by 0.1, but this is unclear to me. An elegant way of associating the two RT distributions to compute KS statistics is provided by Voss, Rothermund & Voss (2004).

Thirdly, because the KS statistic is based on the cumulative probability difference between model and data, for each observed RT, it becomes important to know the predicted

cumulative probability for an observed RT. Given that the likelihood of the model is unknown, it is not clear to me how this was achieved.

I also have some minor points which can be easily addressed:

First sentence of introduction: "one" needs to decide whether "they"...

Page 7. It is discussed that computational models of decision making only produce estimates of mean accumulation rate (and variance), but this is not true. There is a relatively recent trend that aims to provide "single-trial parameter estimates", often by capitalizing on trial-by-trial variability in neural measures (e.g., Gluth & Rieskamp, 2016; Nunez et al., 2015; Turner, Van Maanen, Forstmann, 2014; Van Maanen et al., 2011).

Page 8/11. The leakage parameter is here equated to an "urgency" measure, but the term "urgency" in (perceptual) decision making is more broadly used to indicate a sense of time pressure, and the exact mechanisms of urgency are currently debated. I think it would be better to choose a more neutral term.

Page 11. The grid of k values that is searched to optimize the model fit is bounded at 0 and 0.2, but Supplementary Table 2 mentions values outside of these bounds.

Page 12: "minimize inductive pick-up and participant's safety". I would suggest "minimize inductive pick-up and maximize participant safety".

Page 14: Drop the minus signs before "600ms and 100ms prior to the response".

Caption of Supplementary Figure 1a. "question" should be "equation" .

Caption of Supplementary Figure 2e. "The shaded area depicts the time interval when evidence accumulation occurs". How was this determined?

Replication study. I said previously, I think this replication is important because it shows the robustness of the results. Is it important that in this task participants were uncertain about the value of the choice options, and had to infer these from previous trial outcomes?

Signed,
Leendert van Maanen

Point-by-point reply (our response in blue)

Reviewer #1 (Remarks to the Author):

This is a very exciting paper using simultaneous EEG and fMRI to study evidence accumulation in value-based decision making. The combination of these two techniques helps to overcome each's limitation: poor spatial resolution with EEG, poor temporal resolution with fMRI. The authors utilize an established behavioral paradigm involving choices between foods, though they also appear to replicate the effects with a second learning task. How the brain accumulates and compares evidence in such decisions is a big question and many papers have tackled it using one technique or the other, relying on assumptions about the activity underlying the BOLD signal (with fMRI studies) or conjecturing about the sources of the activity (with EEG studies). In this paper the authors claim to have found a signal, originating in the posterior medial frontal cortex, that shows the expected ramping up of activity in the EEG data. These are certainly exciting findings. What remains unclear though is whether this activity truly reflects value-based evidence accumulation, or some other signal. This, and some other statistical questions, need to be cleared up before I can fully support publication.

We thank the reviewer for the positive assessment of our general experimental approach and the potentially exciting nature of the results as well as the constructive feedback, which we now address in the revised manuscript as detailed below:

(1) The authors fit a sequential sampling model (Ornstein-Uhlenbeck process) to the participants' behavioral data, then compare the model output with EEG activity. However, it appears that the authors, in identifying EEG activity that correlates with the model output, did so using only one EA profile per participant. I don't understand why the authors did it this way, since different trials had different VD and so should have exhibited different average EA curves. Basically, the authors need to do more to convince us that the EEG activity they've identified truly represents evidence accumulation. Other papers in this literature have looked at things like modulation with VD, accumulation to a fixed bound, etc. These are basic tests that should be performed.

We thank the reviewer for bringing this up and we have now performed additional analyses to investigate the response profile of the EA curves in terms of VD (easy vs hard) and response times (fast vs slow) to expose potential modulations along these dimensions. Moreover, we looked at the extent to which the rate of accumulation predicted task performance across participants. Finally, we performed an additional EEG experiment in which new participants passively viewed stimulus pairs used in the original value-based experiment to show that the process of evidence accumulation is present only when a value-based decision is required (i.e. to rule out that the EA curves are driven by low-level perceptual processing of the stimuli as such).

EA modulation as a function of task difficulty and response time.

These findings are now reported in Supplementary Figure 2 and discussed briefly in the main text on pg. 4. In short, we show that there is an average slope modulation along both

dimensions and that ultimately all curves converge to a common boundary near the time of response. On average, fast trials have steeper accumulation rates than slow ones. Correspondingly, easy trials (regardless of RT) also appear to have steeper accumulation rates than difficult ones. We note that these slope modulations are more pronounced along the RT dimension compared to the VD dimension. We posit that this observation is due to the subjective nature of value-based decisions (i.e. the absence of an objective and consistent difficulty level) that ultimately arises from the subjective (and likely noisy) nature of the item preference ratings themselves. This would be in contrast to perceptual decisions, for example, where task difficulty is under precise experimental control, and often spanning a larger range of difficulty levels. These effects are perfectly comparable to those reported in the Polania et al., 2014 Neuron paper (see their supplementary figure 4b).

Accumulation rate predicts performance across subjects.

We also investigated whether the quality of the evidence accumulation (as indexed by the rate of accumulation) predicted individual performance on the value-based task. We hypothesized that individuals with higher accumulation rates would exhibit better performance (higher accuracy) on the task. We therefore computed the slope of the average EA curve for each participant and correlated this with each subject's average performance on the task. We found a significant positive association between the two measures across subjects (Supplementary Figure 2c), further endorsing the notion that our original build-up activity in the EEG is likely to reflect a decision signal, which inform the behavior. We refer to this new analysis in the main text on pg. 4.

Absence of EA during passive viewing.

To provide additional evidence that the EA signals we uncovered correspond to decision-related activity we collected new EEG data during a task in which participants passively viewed pairs of the same stimuli used in the original value-based task (i.e. a decision was no longer required). We show that during these trials the activity on the EA-related sensors we identified in the value task no longer exhibited a ramping up profile (Supplementary Figure 3b), suggesting that our original EA activity likely reflects a true decision-related signal. We refer to this new analysis in the main text on pg. 4-5.

On a related point, the reported correlation here is likely inflated due to the fact that the "best" electrode is chosen for each participant. While this does make sense for then correlating it with the fMRI data, it does not give an accurate representation of the true correlation. The authors could perhaps use their other experiment to estimate the correlation out of sample.

We thank the reviewer for pointing this out and we agree that the correlation appears inflated when considering only the "best" electrodes. We now also compute the average correlation of the EA curve with every electrode in the centroparietal cluster (obtained at the group level) and across participants ($r = 0.68 \pm 0.15$, $t(188) = 30.9$, $P < 0.001$). We note, however, that this approach is likely to underestimate the true correlation, due to

between-subject variability in the exact cluster location (see Supplementary Fig. 1e). For this reason we now report both measures in the main text on pg. 4.

On another related point, it is unclear throughout the manuscript how these reported correlations (and their significance) were calculated. There were obviously repeated observations for each participant. So are these correlations run at the individual level and then averaged? Or was some sort of mixed-effects analysis performed?

We agree with the reviewer that details of these correlation analyses were not described in sufficient detail. Indeed, the original correlations were ran at the individual level (for each subject's best sensor) and then averaged across subjects. We have clarified this in the main text and in the methods. Correspondingly, we also added more information on how we obtained the new correlation results reported on the previous point above.

(2) Why is it “crucial” that the slopes of the accumulating activity are not predictive of RTs (as the authors write near the bottom of p.4)?

We agree with the reviewer that this sentence is unclear and potentially a bit misleading. Indeed what is crucial is not that the slopes are not predictive of the RTs but rather that they are not *strongly* predictive (we only find a moderate correlation between the two), as both the decision process and motor preparation are noisy and add trial-by-trial variability to the response. This is important for our purposes, as it suggests that the electrophysiological variability in the decision process (EA) might offer additional explanatory power in our EEG-informed fMRI regressor (compared to using purely behavioral measures, such as RTs, which are likely to be contaminated by the additional variability in the motor response) in order to expose the relevant accumulating regions. We edited the text to make this point clearer on page 5-6.

The authors seem to be making a distinction between “drift” and “diffusion” and arguing that the drift is being reflected by the EEG activity, but not the diffusion. When they say that the shortest RTs are flat with respect to “rates of evidence accumulation” they mean that RTs are flat with respect to mean drift rate. Of course RTs are NOT flat with drift + diffusion, its just that we don't normally measure the diffusion part. In any case, from looking at Supplementary Figure 2, it appears that RTs are not flat with respect to the EEG activity. Instead they appear to peak when the EEG slope = 0. The authors should try a correlation between absolute EEG slope and RT.

We thank the reviewer for raising this point, as it allows us to clarify that we performed the short/long RT analysis only to confirm that individual RTs cannot be used to reliably index the accumulation process in individual trials (as discussed in the previous point). It was not our intention to interpret this as evidence that RTs are flat with drift+diffusion but merely that the EEG slopes might provide a better estimate of the underlying decision process than RTs. Indeed this analysis only considers drift rates/slopes and only serves as

evidence that the EEG might capture the decision process more reliably than RTs and therefore offer additional explanatory power for the fMRI analysis.

When we move on to the EEG-informed fMRI analysis, however, we do effectively consider the “diffusion” part of the decision process. There, we use the entire trace of the EEG activity, which incorporates the moment-by-moment changes in the accumulation process (i.e. we consider within trial variability), which cannot be captured fully by constant terms such as drift/slope or RT. In fact we view this as the main novelty of our work whereby we use the full temporal dynamics of the decision process to inform the analysis of the fMRI data (rather than average model parameters such as drift rate used in more conventional model-based fMRI). We now highlight this point more explicitly in the discussion on pg. 8.

(3) Given the comments above and the overlap of the pMFC region with the SMA, I have to wonder if the activity that the authors are seeing is instead perhaps related to boundary collapsing or an urgency signal that increases over time, but does not otherwise strongly relate to VD or RT. Given the prior literature linking the SMA and boundary setting, this seems like a rather plausible alternative explanation. I wonder if the authors could address this?

This is an excellent point and we now address this potential alternative interpretation directly by performing an additional analysis on our data and expanding our discussion on this point further. Specifically, we estimated individual trial boundaries based on the EEG traces (i.e. EEG amplitude differences between the onset and offset of accumulation) and we included these estimates as a separate parametric predictor in our fMRI GLM analysis. We found that the activation in the pMFC remained attached to our original EEG regressor that represents the full temporal dynamics of the decision process rather than being absorbed by the new boundary regressor. We formalized this observation by showing that our original regressor was a “significantly” better predictor of the fMRI signal in the pMFC than the boundary regressor ($t(20) = 4.21$, $p < 0.001$; paired t -test comparison of the β coefficients in pMFC). We view these results as evidence that activity in pMFC cannot be explained purely based on boundary adjustments but rather by considering the whole decision process.

Moreover, when comparing long and short RT trials we saw similar results in pMFC (Supplementary Figure 4c-d) suggesting that this region was not modulated by systematic differences in an urgency signal. As a final point, we note that our region differs somewhat from previous reports on boundary adjustment that have focused primarily on the pre-SMA (e.g. as in perceptual decision making) and instead appears to encompass the SMA proper and extend ventrally to incorporate portions of the anterior mid-cingulate cortex.

We now summarize these findings in the main text (pg. 10) and we describe the new analysis in the methods (pg. 19-20).

Minor comments

(4) It should be mentioned that there are alternative views about how evidence accumulation should be represented in BOLD activity. Hare et al. 2011 (PNAS) argue for

the opposite and find a region in dmpfc that this paper also appears to find. This warrants at least some discussion, if not more analyses. Mulder, van Maanen, and Forstmann (2014) provide an interesting discussion of this issue.

We thank the reviewer for pointing us to the Hare et al. paper. We were originally citing this paper for its analysis of task-dependent connectivity, but indeed we should have also referred to it for its discussion on the representation of evidence accumulation in the BOLD signal. We now have corrected this shortcoming in the discussion. However, our take on the Hare paper is that their predictions (their fig. 2D) are in fact in line with what we report here as they suggest that the activity of their comparator region should decrease as the task becomes easier (i.e. as the value difference in the alternatives increases) and the slope of the accumulation increases.

We also agree with the reviewer that Mulder, van Maanen, and Forstmann (2014) provides an interesting discussion on the issue of how evidence accumulation could be represented in the BOLD signal. In our experience the discrepancies can be reconciled by considering differences in the task design. More specifically, diffusion models predict either an Easy > Hard BOLD pattern, assuming that activation remains elevated after reaching the decision threshold (i.e. as in delayed response tasks), or a Hard > Easy pattern, if activation returns to baseline after reaching the threshold (i.e. as in RT-tasks), since low evidence accumulates slower than high evidence, leading to greater integrated activity. Our current results are in line with the latter interpretation. We briefly touch on this point and reference the Mulder et al., paper in the discussion (pg. 5).

(5) What is the rationale for the relative weighting of choice and RT data in the LL function? Were there 5000 simulations per VD, or overall?

We agree with the reviewer that these parameters were not properly explained. We have made it clear in the methods that the 5,000 simulations were per VD level. The relative weighting of accuracy and RT in the LL function was chosen in order to make the scaling of the two measures more comparable. This scaling procedure appeared to improve the accuracy fits compared to initial simulations with equal weight on the two variables.

(6) Analyses using RT as the dependent variable should use $\log(\text{RT})$ instead, since RTs are typically log-normal.

We thank the reviewer for suggesting this, we have now rerun the correlation using $\log(\text{RT})$ instead of RT. The results are consistent with what we were already reporting (slopes are only partially correlated with RTs and more so for long RTs) and are shown in the updated Supplementary Figure 4a.

(7) Was there any cost of a “lost trial” to the participants?

No, there was no cost for lost trials and overall these were extremely rare (<1% of all trials). We now state this explicitly in the manuscript.

(8) There are a lot of technical details when relating the EEG data to the fMRI data. This is obviously a critical contribution of the paper, so it would be helpful to point what here has been established already and what methods are novel to this paper.

We feel that much of the novelty of our approach stems from the fact that we are exploiting electrophysiological variability (within and across trials) to inform the analysis of the fMRI data. While we use a computational model to select and constrain which features (electrodes, time window) of the EEG data we consider relevant for the fMRI analysis, our approach differs from conventional model-based fMRI in that we use the full temporal dynamics of the decision process (rather than average model parameters such as drift rate or boundary) to inform the analysis of the fMRI data. As such, we don't make any a priori assumptions about which characteristic of the EEG response is relevant (e.g. the slope or the boundary of the accumulation) but rather consider the moment-by-moment changes in the EA time-series in the selected electrodes to capture *all* relevant variability, which could possibly explain the fMRI signal. Overall, this approach offered more predictive power over and above what could already be conferred by behavioral measures or other model parameters (e.g. as in the boundary analysis above). We now discuss this point explicitly in the discussion (pg. 8-9).

Reviewer #2 (Remarks to the Author):

I have previously reviewed this manuscript at a different journal and, while many of the original reviewer comments have been addressed, some important ones have not and so they are repeated below here.

In this paper Philiastides et al seek to shed light on the neural origins of value-based decision making through the combination of computational modelling and the fusion of simultaneously recorded EEG and fMRI data. Participants completed a two-alternative relative value judgment task. The authors fit a leaky accumulator model to the behavioural data and then identified EEG electrodes whose trial-averaged response-aligned activity exhibited a significant correlation with the model-predicted evidence accumulation time-courses. Single-trial activity from these electrodes was then included as a regressor in the fMRI analysis and this approach highlighted the posterior medial frontal cortex. The authors are also able to demonstrate that this same region exhibits negative coupling with ventromedial prefrontal and orbitofrontal cortices- two regions previously implicated in representing value. The authors conclude that the pMFC likely plays a role in motor control and thus represents emerging decisions in a generic fashion, not limited to value-based decisions.

This is an interesting and very nicely written manuscript that brings a new and potentially valuable methodology to bear on an important question. However, I am still not convinced that the strong claims made by the authors are warranted based on these data.

We thank the reviewer for the overall positive assessment of the manuscript and for highlighting the potentially valuable contribution of this work. The reviewer's remaining comments prompted us to collect additional data and perform new analyses to provide further support for our claims as we explain below:

Major Comments:

The significance of the conclusion offered by the present study rests on the idea that there might be a general-purpose accumulator that, just as it accumulates perceptual information for perceptual decisions, accumulates value information for value-based decisions. But the alternative choice items are presented visually, and subjects have to identify the items in order to know their value and compare them. So how can it be determined that any observed buildup dynamics correspond to the accumulation of the value information and not the perceptual information that is required for the more primary task of figuring out what one is looking at?

We thank the reviewer for this very constructive comment. We agree that one should investigate further whether the perceptual processing of the stimuli themselves does not drive the build-up dynamics we observe in the data. For this reason we ran a new EEG experiment in which participants passively viewed pairs of the same stimuli used in the original value-based task (i.e. a task in which a decision was no longer required). We show that during these trials the activity on the sensors that exhibited accumulation dynamics in the value task no longer exhibited a ramping up profile (Supplementary Figure 3b). Moreover, early perceptual encoding of the stimuli as such usually occurs within the first couple of hundred milliseconds (seen as the early EEG evoked responses in Supplementary Figure 3a-d), well in advance of the accumulation-like dynamics seen in the value-based task. Taken together, these findings suggest that our original evidence accumulation activity likely reflects a true decision-related signal pertaining to value choices. We refer to this analysis in the main text on pg. 4.

With this in mind it is actually very surprising that only a single brain region is found to correlate with the EEG accumulator activity – where are the perceptual decision regions highlighted in previous work? A recent study by Polania et al (2014) did succeed in dissociating EEG markers of perceptual vs value-based decision making yet these are not considered for analysis here.

In our experiment the value-based decisions (which item one prefers to eat) are likely orthogonal to any potential perceptual differences in the stimuli (e.g. which item is bigger). In other words, over hundreds of trials, items and subjects, the item preferences are likely decoupled from subtle differences in the perceptual properties of the items. To expose perceptual decision making activity, the authors in the Polania et al. paper used a separate perceptual decision making task, whereby they asked participants to judge the size of the presented items, rather than choose based on their preference for the items.

Motivated by the reviewer's comments and the earlier work by Polania et al., we ran a separate EEG experiment in which subjects were presented with pairs of the same stimuli used in our original value-based task but they were instead asked to indicate which item they thought was larger in size (i.e. perform a series of perceptual judgments). We found that the same centroparietal EEG sensors, which exhibited accumulation dynamics during the value-based task, also showed a comparable build-up of activity during perceptual decision making (Supplementary Figure 3c). This finding is in perfect agreement with the Polania et al. work (i.e. accumulation dynamics on centroparietal electrodes for both value and perceptual judgments), which we view as support for the notion that the process of evidence accumulation in pMFC might constitute a general-purpose decision mechanism.

We now refer to these findings in the main text (pg. 9) but nonetheless acknowledge that for perceptual decisions additional brain networks (such as IPS, dlPFC) are likely to be involved as was shown in previous work.

The title of this paper and the abstract both strongly emphasise the role of pMFC in value-based decision making yet the authors have no evidence that the pMFC's role is specific to valuation and in fact conclude in the Discussion that it most likely plays a generic role in decision making. I think this is a highly plausible interpretation but, without inclusion of a perceptual decision task and a condition in which participants used an alternative effector to indicate their decisions, it is something of a speculation. Moreover, this interpretation of pMFC as a generic motor control region calls into question the emphasis in both the title and abstract that the pMFC is the site of value-based decision making which implies that the study has identified a region that is specifically involved in accumulating value. The authors should also consider in more detail the previous extensive fMRI literature on decision making and whether or not the pMFC has been highlighted previously (e.g. Liu & Pleskac J Neurophysiol 2011).

We thank the reviewer for this point, and we have acted upon it in two ways:

1. We ran a separate EEG experiment in which we asked participants to perform perceptual judgements using pairs of the same stimuli used in our original value-based task by asking them to report which of the two stimuli was bigger in size. All aspects of the experiment (stimulus duration, event timing, EEG equipment etc) were otherwise identical to those used in the original experiments. We found that the same centroparietal EEG sensors, which exhibited accumulation dynamics during our value- and reward-based tasks, also showed a comparable build-up of activity during perceptual decision making (Supplementary Figure 3c). We view this finding as evidence that information accumulation is a general decision mechanism and that pMFC might be a common module of the network driving such a process. Critically,

this doesn't rule out the involvement of other regions that have consistently been implicated in perceptual decision making in the past (such as IPS, dlPFC). We also discuss these results in the main text on pg. 9.

2. We have toned down the concluding remark of the paper, to make it clear that even though our data support the idea that pMFC acts as an accumulator region for value- and reward-based decisions this doesn't imply that its role is specific to valuation. Instead we discuss the possibility that it is also involved with decisions in other domains (e.g. perceptual decision making) as suggested by our new experiments. We are now also more explicit that we chose to focus primarily on value- and reward-based decisions as efforts to identify potential accumulator regions in this domain received considerably less attention than in the long body of literature in perceptual decision making (see pg. 10, where we also discuss the role of pMFC across domains, including the previous work by Liu & Pleskac 2011).

In conclusion, we agree that to offer concrete evidence that the pMFC is a general-purpose decision maker, one needs to test a larger set of tasks and effector modalities. However, collecting simultaneous EEG/fMRI on such a wide range of tasks and conditions is beyond the scope of this work and will prevent us from showcasing the results on value-based accumulation (which we feel is the main novelty of this work) in a timely manner. We hope that the additional EEG experiments and analyses we performed here along with the changes we made to the manuscript help paint a more accurate picture of the main contributions of this work.

The authors highlight that 'the slopes of this accumulating activity were not strongly predictive of RTs due to the high degree of inter-trial variability in decision processing.' The authors should clarify what they mean by 'decision processing' but a graver concern is that they go on to include RT as an additional regressor of no interest in the fMRI analysis. Even if the influence of accumulator build-up rate on RT is weak for early trials, how is the cross-trial variation in build-up rate at all meaningful if it can have no impact on behaviour? The inclusion of evidence strength as a regressor-of-no-interest is equally problematic since an accumulator process should also covary with this variable. The analysis is constructed in a way that would systematically tend to remove true evidence accumulation processes from the picture. This might partly account for the rather surprising fact that only one brain region appears to be associated with decision formation on this task.

We generally agree with the above argument and ran another GLM where we removed the two regressors for the RTs and the strength of the evidence and the results remained unchanged. More specifically, the only activation we see in the EEG-informed regressor capturing accumulation dynamics was in the pMFC. Statistical significance of this region improved somewhat but no new areas appeared. Regions previously absorbed by the RT regressor moved to our constant term regressor (i.e. our unmodulated regressor during decision making). This finding suggests that it is truly the endogenous electrophysiological variability (in our EEG regressor) that is driving the observed effect. We now report the details of this analysis and the associated results on pg. 19.

We originally included separate regressors for these other quantities because they also correlate strongly with the attentional demands of the task and stimulus difficulty/salience (and as such are often seen and discussed in the literature as confounding variables). Correspondingly, they mainly absorb activity from regions of the human attentional network (Supplementary Table 1), which might exert an influence on but are not part of the core decision making (accumulator) network as such. In other words, our main goal was to expose regions that continued to reflect endogenous variability in the evidence accumulation process after stimulus and task difficulty have been accounted for (i.e.

expose variability across (nominally) equally difficult trials). We now make this point clearer when we present the original GLM design in the methods section on pg. 19.

Finally, we clarify that when we talk about “variability in decision processing” we refer to the endogenous variability in all relevant decision parameters, including starting point, accumulation rate and boundary, which are likely to be affected by the stochastic nature (noise) of the decision process on individual trials. Correspondingly, when we exploit the EEG signal related to the decision process, we are not making any a priori assumptions about which characteristic of the EEG response is relevant but rather consider the moment-by-moment changes in the EA time-series in the selected electrodes to capture *all* relevant variability, which could possibly explain the fMRI signal. We now state this explicitly in the discussion on pg. 8.

I also have a concern about the slope measurements. A window starting 600ms before response execution was used which means that for any $RTs < 600ms$ the trial will incorporate activity that precedes stimulus onset. Surely this is problematic and would also lead to meaningless slope measurements on certain trials? Here the paper would benefit from some 'sanity check' analyses such as demonstrating that the build-up rate of the EEG activity is faster for easy vs difficult trials.

These are all excellent points, which we have now carefully addressed. As we explain below in detail, we recomputed individual trial slopes with more appropriate window definitions and repeated the relevant analyses as well as performed several additional sanity checks to confirm that the build-up of activity we see in the EEG is indeed related to information accumulation (including, among other things, the easy/hard analysis the reviewer suggested).

Individual trial slopes:

We now compute single trial slopes on trial-specific windows consisting of the entire trial duration (i.e. RT) minus the subject specific non-decision time nDT estimated by the diffusion model and an additional 100 ms to account for motor preparatory activity. This ensured that slope measurements were performed only during the decision period and in all cases never incorporated activity that preceded stimulus onset. We report this approach on pg. 19. We repeated all analyses that relied on these individual trial slopes (i.e. correlations between slopes and RTs, GLMs for the analysis of the fMRI data) and found that our results remained unchanged. Finally, we note that the window from 600 to 100 ms before response is now only used in the correlation between average response-triggered EEG traces and the evidence accumulation (EA) profile predicted by the model to identify sensors of interest (after taking into account that the average reaction time was in the order of ~ 800 ms).

Slope modulation as a function of task difficulty and response time.

We ran additional analyses to show that there is an average slope modulation along both dimensions and that ultimately all curves converge to a common boundary near the time of response. On average, fast trials have steeper accumulation rates than slow ones. Correspondingly, easy trials (regardless of RT) also appear to have steeper accumulation rates than difficult ones. We note that these slope modulations are more pronounced along the RT dimension compared to the task difficulty dimension. We posit that this observation is due to the subjective nature of value-based decisions (i.e. the absence of an objective and consistent difficulty level) that ultimately arises from the subjective (and likely noisy) nature of the item preference ratings themselves. This would be in contrast to

perceptual decisions, for example, where task difficulty is under precise experimental control, and often spanning a larger range of difficulty levels. These effects are perfectly comparable to those reported in the Polania et al., 2014 Neuron paper (see their supplementary Suppl. Fig. 4b). These findings are now reported in Supplementary Figure 2 and discussed briefly in the main text on pg. 4.

Accumulation rate predicts performance across subjects.

We also investigated whether the quality of the evidence accumulation (as indexed by the rate of accumulation) predicted individual performance on the value-based task. We hypothesized that individuals with higher accumulation rates would exhibit better performance (higher accuracy) on the task. We therefore computed the slope of the average EA curve for each participant and correlated this with each subject's average performance on the task. We found a significant positive association between the two measures across subjects (Supplementary Figure 2c), further endorsing the notion that our original build-up activity in the EEG is likely to reflect a decision signal, which inform the behavior. We refer to this new analysis in the main text on pg. 4.

Reviewer #3 (Remarks to the Author):

The authors report a study in which BOLD fMRI is predicted using EEG signatures of evidence accumulation. They find that BOLD responses in pMFC reflect a rising (accumulating) signal in a cluster of centroparietally located electrodes. These electrodes were identified earlier by studying the correlation between the hypothesized evidence accumulation signal and the EEG signal. The authors conclude that the results suggest that activity in pMFC reflects a decision formation, contrary to earlier reports that these areas (or areas in close proximity) reflect a decision bound (e.g., Ivanoff et al., 2008; Forstmann et al., 2008, Van Maanen et al., 2011; Van Veen et al., 2008; Winkel et al., 2012). If this conclusion is valid, this finding is exciting and might encourage all of us to think about decision making processes in a lot less static way, as it questions the often assumed dichotomy between “evidence accumulation areas” and “decision bound areas”.

From a methodological view point there are many aspects of this paper that I like, but I also have a number of methodological concerns. First of all, I think the independent replication is a strong asset of this study, as it shows the robustness of the results, which is often problematic in (model-based) fMRI studies (e.g., Button et al., 2013). Secondly, I appreciate that the authors chose to include the EEG signal itself as a regressor for the fMRI analyses, rather than a model-based estimate of the evidence accumulation process. This way, a potential misspecification of the evidence accumulation model has less influence on the eventual inference.

We thank the reviewer for the supportive as well as constructive comments, and we hope that the improvements we have made to the paper will further support the original findings.

My main concerns all relate to the computational modelling part of the paper, and can be divided in two broad categories:

1. The choice of the leaky accumulator model.

An important aspect in this study is the specification of the evidence accumulation model, as it directly influences the choice of electrode for the fMRI analyses. The authors opted to fit a leaky accumulator model. The inclusion of the leak term is however not supported. Wouldn't a model without leak be a better (more parsimonious) choice? Figure 1c suggests that the normalized EEG activity ramps in a more or less linear fashion. This could be due to averaging across subjects of course, but it could also reflect that the evidence fully integrates (ie., no leak). Another more practical benefit of a model without leak term is that estimating parameters is less error prone (more on this under point 2).

A second choice in the model that I think deserves more scrutiny is the choice to constrain the parameter scaling by fixing the thresholds at 1 and -1. This constraint embodies the strong assumption that individual variability in choice behavior is fully explained by the evidence accumulation process, and not by variability in threshold setting. This is important given the alternative hypothesis that pMFC reflects a decision bound (as eluded to above). If the evidence accumulation parameters (λ and k) absorb variability in choice behavior that is due to individual differences in threshold, then a model that allows these differences might explain the data better, and might alter the interpretation of these findings.

We thank the reviewer for this thoughtful comment, which we now address by better highlighting some important aspects of our data and by performing a series of new analyses as highlighted below. The reviewer's intuition is right in that the seemingly linear ramping of the EEG signal in Fig.1C is likely the result of averaging across subjects and across trials. Our initial choice for the leak term was motivated by the observation that both single EEG trials and single subject averages do indeed show evidence of a non-linear growth, as can be seen in the representative examples in Supplementary Figure 1d and Supplementary Figure 4e. We nonetheless agree that we should have formally tested (and compared) a model in which the boundary was treated as a free parameter.

In doing so we used two new model parameterizations where we introduced a threshold parameter θ to potentially account for between subject variability in the decision boundary, as suggested by the reviewer. In the first new model we substituted the leak parameter λ with a threshold parameter θ ('boundary' model), while in the second we used both leak and boundary parameters ('joint' model). The remaining parameters stayed the same as in our original model.

Specifically, we ran the new models on the same parameter intervals we used for k , σ , nDT (and λ for the 'joint' model) in our original model, while we allowed the threshold parameter θ to vary in the range $[0.5:1.5]$. Overall, the new models did capture the main behavioral effect in terms of RTs and accuracy, but they did not seem to fit the data better than the leaky accumulator model we used previously (Supplementary Figure 6a). When we formally compared the performance of the two new models against the old one using the Bayesian Information Criterion (BIC) the leaky model marginally outperformed both the 'boundary' model and the 'joint' model (BIC from leaky model: 29.45 BIC from 'boundary' model: 32.87, BIC from 'joint' model: 34.96). Expanding the range for k and σ further in the new models failed to improve the fits further. We now report these analyses on pg.14 of the main text and in Supplementary Methods. Finally, we compared the evidence accumulation (EA) curves produced by all models against the EEG data (see new Supplementary Figure 6b), which further highlights that our original model remained a better fit to the neural data. Overall, we suspect that our leak term might – implicitly – be absorbing (among other things) some variability associated with boundary adjustments. Therefore as a more direct way of testing whether activity in the pMFC is driven instead only by boundary adjustments we performed a separate analysis.

Specifically, we estimated individual trial boundaries directly from the EEG traces (i.e. EEG amplitude differences between the onset and offset of accumulation) and we included these estimates as a separate parametric predictor in our fMRI GLM analysis. We found that the activation in the pMFC remained attached to our original EEG regressor that captures the full temporal dynamics of the decision process rather than being absorbed by the new boundary regressor. We formalized this observation by showing that our original regressor was a "significantly" better predictor of the fMRI signal in the pMFC than the boundary regressor ($t(20) = 4.21$, $p < 0.001$; paired t -test comparison of the β coefficients in pMFC). We view these results as evidence that activity in pMFC cannot be explained purely based on boundary adjustments but rather by considering the whole decision process. In other words, it is the full temporal dynamics of the decision process (i.e. moment-by-moment fluctuations in the accumulation process) that better explain

activity in the pMFC. We now present this new analysis on pg. 10 in the main text and in Supplementary Figure 4e.

2. The parameter estimation of the accumulator model.

The authors' choice of model requires a simulation of the predicted RT distributions during the parameter optimization procedure, because there exists no closed-form likelihood function of this particular model. Such a simulation may lead to misspecification of parameters, because of noise in the predicted RT distributions. A recent study in my lab (currently under review, but I can send a draft of the manuscript if desired) shows that in particular the leaky competitive accumulator model is prone to this, partly because of strong tradeoffs between the estimates of k and λ (and inhibition, but that is not included in the current model). The correlation of 0.60 between the λ and k parameter presented in Supplementary Table 2 hints that this may be a problem here as well. It could very well be that this is a minor issue in this study, because of the nice constraint on the k parameter (the estimated trial difficulty using individual value estimates of all presented items). Nevertheless, it would be good to see a parameter recovery analysis.

We thank the reviewer for highlighting this potential issue with the parameter estimation in our model. To address this issue we ran a parameter recovery analysis as suggested by the reviewer (in a similar manner as in Miletic et al. J. Math Psych 2016, which we now cite). We generated one simulated behavioral data set (i.e. full RT distributions for correct and incorrect trials) by running 5000 simulations of the SSM for each VD level using the average parameters estimated originally on the real behavioral data (i.e. as in Supplementary Table 2). Additionally we generated five more simulated behavioral data sets using five randomly sampled parameter sets from the range used in the original fit. For each simulated behavioral data set we ran the SSM this time trying to fit the RT distributions and identify the set of model parameters that maximized our KS statistic in the same way we did for original behavioral data. To assess the recoverability of our parameters we repeated this procedure 10 times for each simulated data set (i.e. 60 repetitions). The recoverability of the parameters of our SSM was high in almost all cases as can be seen in Supplementary Figure 6c-f. We now summarize this procedure and the results in the Methods section in the main text (pg. 14) and in the supplement.

In addition, I am not convinced that the statistic that is optimized yields the best possible set of parameters. Firstly, I think that the KS statistic does not represent the likelihood of the data under the model, but rather the probability that two distributions are identical. While these two concepts are related, I think it would be good to be precise about what is being optimized.

We thank the reviewer for pointing this out. What we are maximizing is indeed the probability that the RT distributions predicted by the model match the ones observed behaviorally. As this is only an approximation of the true likelihood function we edited the formula and the text on page 13 to state this explicitly.

Secondly, the weighting of the different components of the KS statistic seems a bit arbitrary. Effectively, the influence of the RT distribution of incorrect responses on the

summed log KS value is assumed as strong as the RT distribution of correct responses, even though typically there are fewer incorrect responses, and therefore the estimate of this distribution is poorer. Similarly, the estimation of accuracy (one data point per participant/condition) is assumed as important as the estimation of a full RT distribution. Perhaps the weight of the accuracy is scaled by the division of the difference in error rates between data and model by 0.1, but this is unclear to me. An elegant way of associating the two RT distributions to compute KS statistics is provided by Voss, Rothermund & Voss (2004).

We thank the reviewer for raising this point and we agree that our choice of weighting the different components of the KS statistic was somewhat ad hoc. We now use the elegant procedure described in Voss, Rothermund & Voss (2004) instead and refit the model. We also note that the model comparisons reported earlier showing a better performance for the leak-model were also done with this new procedure. We have, therefore, recomputed the optimal parameters for every subject (updating Supplementary Table 2) with this new measure. We found that for the large majority of subjects the selection of the best EEG sensor to be used in the fMRI analysis remained unchanged. For 4 subjects the best sensor changed slightly but in all cases remained within the original centroparietal electrode cluster. As a result we repeated all the analyses (updating the relevant figures) and found that the results remained virtually unchanged. In fact, our original cluster in the pMFC became statistically even more robust (updated Figure 2c).

Finally, we note that the relative weighting of accuracy and RT in the LL function was chosen in order to make the scaling of the two measures more comparable. This scaling procedure appeared to improve the accuracy fits compared to initial simulations with equal weight on the two variables.

Thirdly, because the KS statistic is based on the cumulative probability difference between model and data, for each observed RT, it becomes important to know the predicted cumulative probability for an observed RT. Given that the likelihood of the model is unknown, it is not clear to me how this was achieved.

We thank the reviewer for this request. We computed the KS statistics with the MATLAB function *kstest2* which estimates the predicted cumulative probability for any observed RT through the proportions of the predicted RTs which are less than or equal to the observed RT. We now explain this in the text more clearly on page 13.

I also have some minor points which can be easily addressed:

First sentence of introduction: "one" needs to decide whether "they"...

We used a singular *they* in this case (i.e. as a gender-neutral singular pronoun).

Page 7. It is discussed that computational models of decision making only produce estimates of mean accumulation rate (and variance), but this is not true. There is a relatively recent trend that aims to provide "single-trial parameter estimates", often by capitalizing on trial-by-trial variability in neural measures (e.g., Gluth & Rieskamp, 2016; Nunez et al., 2015; Turner, Van Maanen, Forstmann, 2014; Van Maanen et al., 2011).

We thank the reviewer for pointing out these papers. We now amended the sentence on page 8, to highlight this new trend and reference the suggested papers in this context.

Page 8/11. The leakage parameter is here equated to an “urgency” measure, but the term “urgency” in (perceptual) decision making is more broadly used to indicate a sense of time pressure, and the exact mechanisms of urgency are currently debated. I think it would be better to choose a more neutral term.

We now use the more neutral expression “acceleration to threshold”.

Page 11. The grid of k values that is searched to optimize the model fit is bounded at 0 and 0.2, but Supplementary Table 2 mentions values outside of these bounds.

We thank the reviewer for spotting this discrepancy. Indeed, the original upper bound of the grid search for k was 0.4 and not 0.2 (that was a typo). We now corrected this problem.

Page 12: “minimize inductive pick-up and participant’s safety”. I would suggest “minimize inductive pick-up and maximize participant safety”.

Done.

Page 14: Drop the minus signs before “600ms and 100ms prior to the response”.

Done.

Caption of Supplementary Figure 1a. “question” should be “equation” .

Done.

Caption of Supplementary Figure 2e. “The shaded area depicts the time interval when evidence accumulation occurs”. How was this determined?

We describe this procedure in the EEG data analysis section in the methods and we added a direct reference to this in the figure caption (original Suppl. Figure 2e is now Suppl. Figure 5b). In short, this interval was selected to span the period between the average onset of accumulation (600 ms prior to the response) and lasting until 100 ms before the response. We purposely excluded the last 100 ms leading up to the response to avoid potential confounds with activity related to motor execution. This is the same procedure adopted by Polania et al., *Neuron*, 2014.

Replication study. I said previously, I think this replication is important because it shows the robustness of the results. Is it important that in this task participants were uncertain about the value of the choice options, and had to infer these from previous trial outcomes?

We thank the reviewer for raising this point. We agree that participants were uncertain about the value of the options but we don’t think this hampers the validity of our analysis for two reasons. 1) Even the presence of uncertainty participants have to assign value to the alternatives on every trial (this is thought to occur through a reinforcement learning

mechanism) and hence use this information to make a decision and 2) in the course of the task participants quickly learn to correctly identify the relative value of the stimuli and choose accordingly (Supplementary Note 1 in Fouragnan et al. 2015). In all cases, we would expect uncertainty about the items value to increase the noise in the decision processing and therefore, if anything, reduce and not inflate the relationship between its EEG and fMRI correlates.

Signed,
Leendert van Maanen

Reviewers' comments:

Reviewer #1 (Remarks to the Author):

The authors did an excellent job with the revision and addressed all of my concerns with new data and new analyses.

I have just one issue that I think the authors need to correct before final publication. One drawback of sequential sampling models, such as the one used here, is that it is not possible to identify the drift rate, noise, and boundaries simultaneously, due to the arbitrary units of EA. In one of the new analyses, the authors add this parameter "theta" to allow the boundaries to vary, in addition to the already varying k and σ . Unless I'm mistaken, there is no way that this parameter can be identified. Therefore, it is no surprise that this more complex model does not work as well as the original model.

To be clear, I do not think this analysis was necessary. The results from Supplementary Figure 4e I think adequately address the issue of evidence accumulation vs. boundary separation. My recommendation would be to omit the "theta" model analyses entirely.

Reviewer #2 (Remarks to the Author):

The authors have gone to great lengths to answer all of the reviewers' questions by collecting additional data and conducting numerous additional analyses. While several of my concerns have now been addressed I am afraid that some of the key ones still stand.

First, I raised the concern that we cannot be sure that the study design potentially confounds the perceptual analysis of stimuli with the value-based decision and this clouds any interpretation of the pMFC activations. The authors respond that the value and perceptual decisions are likely orthogonal on this task but I am not convinced. In order to judge the relative value of the stimuli you must first identify them and we do not know how long this would take in the present experiment. The authors conducted a control experiment in which participants compared the sizes of two stimuli and that this highlighted the expected accumulator dynamics in the centro-parietal. I very much appreciate the trouble and effort the authors have gone to here but my query relates to pMFC not the centro-parietal EEG cluster. Can the authors rule out that pMFC also shows activation during purely perceptual decisions? I note too that size comparison would be a far simpler and quicker decision than establishing the identity of the two items

My second concern again relates to the fact that, as acknowledged by the authors, the role of the pMFC may well be a domain-general one, not specific to value based decisions. This calls in to question the novelty of this particular paper then in the context of a fairly extensive series of fMRI papers that have sought to identify domain-general decision regions and, due to the implementation of differing methodologies and analytic assumptions, have highlighted a disparate set of brain regions. So really here I am pondering the highly subjective matter of whether the results are sufficiently new and

definitive to warrant publication in this particular journal.

Reviewer #3 (Remarks to the Author):

I would like to thank the authors for performing the additional analyses, and I am happy to see that the results remain, and if anything are even stronger.

Three relatively small issues remain in my opinion that could be addressed, related to my earlier points:

1. The authors fit additional models to test alternative hypotheses, but it seems they did not include the simpler model that only estimates drift, diffusion and NDT (ie neither leak nor threshold variability across participants). I think it would be good to exclude this model as well.
2. How was the model comparison performed? Specifically, are the reported BIC values summed over participants?
3. Supplementary figure 6a seems to suggest that the joint model fits actually worse than the leak model. Given that these are nested models, this should not be the case. The easiest solution to this issue would be to use the best fitting parameters of the leak model as initial estimates of the joint model, and show that individual estimates of the threshold parameter do not improve the fit.

Congratulations on this great work!

Signed,
Leendert van Maanen

Point-by-point reply (our response in blue)

Reviewer #1 (Remarks to the Author):

The authors did an excellent job with the revision and addressed all of my concerns with new data and new analyses.

We thank the reviewer for the positive assessment of our work and for the constructive feedback in the previous round of reviews – which we feel has helped improve the manuscript greatly.

I have just one issue that I think the authors need to correct before final publication. One drawback of sequential sampling models, such as the one used here, is that it is not possible to identify the drift rate, noise, and boundaries simultaneously, due to the arbitrary units of EA. In one of the new analyses, the authors add this parameter "theta" to allow the boundaries to vary, in addition to the already varying k and σ . Unless I'm mistaken, there is no way that this parameter can be identified. Therefore, it is no surprise that this more complex model does not work as well as the original model.

To be clear, I do not think this analysis was necessary. The results from Supplementary Figure 4e I think adequately address the issue of evidence accumulation vs. boundary separation. My recommendation would be to omit the "theta" model analyses entirely.

We agree with the reviewer's intuition on the ability to recover the new "theta" parameter and that the new analysis in Fig. 4e is sufficient to address the issue of evidence accumulation vs. boundary separation. We initially introduced the "theta" model in response to a comment from another reviewer requesting to investigate the possibility that "individual variability in choice behavior" could be explained "by variability in threshold setting". Therefore, while we agree with the reviewer that the "theta" parameter is difficult to identify together with drift rate and noise, we have included it in the supplement to provide partial control for subject-to-subject variability in decision boundary, as requested by R3. We have nevertheless emphasized in the text (pg. 15) that the identifiability issue is the likely cause of the lack of improvement in the fit by the new model. Similarly, we now stress explicitly the importance of using the EEG signal itself as a regressor for the fMRI analysis (rather than the model-based estimate of the evidence accumulation). This way a potential misspecification in the model has a lesser influence on the eventual inference (we view this, in fact, as one of the novel contributions of the work).

Reviewer #2 (Remarks to the Author):

The authors have gone to great lengths to answer all of the reviewers' questions by collecting additional data and conducting numerous additional analyses. While several of my concerns have now been addressed I am afraid that some of the key ones still stand.

We thank the reviewer for the additional feedback, which we address below.

First, I raised the concern that we cannot be sure that the study design potentially confounds the perceptual analysis of stimuli with the value-based decision and this clouds any interpretation of the pMFC activations. The authors respond that the value and perceptual decisions are likely orthogonal on this task but I am not convinced. In order to judge the relative value of the stimuli you must first identify them and we do not know how long this would take in the present experiment. The authors conducted a control experiment in which participants compared the sizes of two stimuli and that this highlighted the expected accumulator dynamics in the centro-parietal. I very much appreciate the trouble and effort the authors have gone to here but my query elates to pMFC not the centro-parietal EEG cluster. Can the authors rule out that pMFC also shows activation during purely perceptual decisions? I note too that size comparison would be a far simpler and quicker decision than establishing the identity of the two items.

In our previous reply we mentioned the orthogonality between the value and the perceptual content of our stimuli (which we now formally quantify and report on pg. 12) only to justify the lack of significant correlation between known perceptual decision regions and our EEG regressor. We agree with the reviewer that in order to judge the value of an item one first needs to identify the stimulus. Correspondingly, we appreciate the reviewer's concern about the potential interference of the perceptual analysis of the stimuli on the process of value evidence accumulation that follows.

However, we believe that the control analyses we performed in the last round of revisions already showed that the value-based decision is largely decoupled from stimulus processing and identification by providing accurate timing information on the perceptual analysis of the stimuli that clearly precedes the process of evidence accumulation that we reported in the paper.

In the first control experiment we asked new participants to passively view the stimuli used in the original value-based task and were explicitly instructed to pay attention to the snack items. This allowed us to study 1) the timing of the early perceptual analysis of the stimuli in the absence of a decision and 2) the extent to which the accumulating activity in the value task was confounded by these earlier stages of processing.

We showed that early stimulus processing is completed within the first 200-250 ms after stimulus presentation (see evoked responses in Supplementary Fig. 3d), well before the accumulating activity which we used in our subsequent EEG-informed fMRI analysis (i.e. no mixing of perceptual and decision signals). This timing is also consistent with the predictions of our computational model (c.f. non-decision time parameter) and the large body of previous literature on the timing of object recognition (Liu et al., Neuron 2009; Bienek et al., EPN 2015). More critically the accumulating activity was completely abolished during passive

viewing, suggesting that it is indeed specific to the decision process itself (see Supplementary Fig. 3b). The evoked responses associated with the perceptual analysis of the stimuli on the other hand were remarkably similar across the passive viewing task and the active value-based and perceptual decision tasks (the latter introduced in a separate control experiment that involved an active comparison of stimulus evidence, Supplementary Figure 3c-d). This finding suggests that evidence accumulation can be decoupled from the perceptual analysis of the stimuli and that the latter appears to have occurred even in the passive viewing condition.

Indeed we made multiple additional efforts to exclude the perceptual response to the stimulus from the EEG-informed fMRI analysis. Firstly, we identified the EEG electrodes that we used to build the regressor through correlations with the *model* EA predictions that by design produce a “pure” decision signal, driven exclusively by value evidence (where early perceptual processing plays no part). Given the high correlations of individual EEG electrodes to these EA model predictions it stands to reason that our EEG activity is largely decoupled from early sensory influences.

Moreover, we actively removed the early perceptual response by excluding the first part of each trial from the raw EEG trace we used to build the regressor. In order to do that we considered the entire trial duration (i.e. RT) minus the subject specific non-decision time *nDT* estimated by the model, which accounted for stimulus processing and motor execution. More specifically, we split this non-decision time in two intervals by fixing the motor preparation to 100 ms prior to the response (when a sudden increase in corticospinal excitability occurs as in Chen et al, 1998) and setting the average duration of the stimulus encoding to the *nDT*-100 ms after stimulus onset (Fig. 2a). This choice was consistent with the difference between the average *nDT* of 290 ms and the initial perceptual modulation in the average EEG response in centro-parietal electrodes lasting around 200 ms (Suppl. Fig. 3d). Taken together, these constraints ensured that the cluster we found in pMFC was strictly linked to the accumulating activity observed in the centro-parietal EEG signal and was largely decoupled from the initial perceptual analysis of the stimuli.

Thus, although we agree with the reviewer that pMFC should not be active during the initial processing and identification of the stimuli (this is known to take place in the ventral visual stream instead), we cannot rule out a potential involvement of the pMFC in the actual process of evidence accumulation itself. There is a fundamental distinction between the passive perceptual analysis of the stimuli and the active process of making a perceptual decision about them (e.g. which item is bigger in size). In the latter case the pMFC might very well be involved if this region is a domain-general decision maker, as suggested by our other control EEG experiment showing accumulation dynamics during perceptual choices in the same electrodes as in our value-based task (Suppl. Fig. 3c).

Finally, we note that our pMFC connectivity analysis revealed unique and task-depend coupling with vmPFC and STR, regions that are known to encode exclusively for value (not perceptual) information, consistent with our interpretation of pMFC encoding value-based evidence accumulation in our task.

My second concern again relates to the fact that, as acknowledged by the authors, the role of the pMFC may well be a domain-general one, not specific to value based decisions. This calls in to question the novelty of this particular paper then in the context of a fairly extensive series of fMRI papers that have sought to identify domain-general decision regions and, due to the implementation of differing methodologies and analytic assumptions, have highlighted a disparate set of brain regions. So really here I am pondering the highly subjective matter of whether the results are sufficiently new and definitive to warrant publication in this particular journal.

If we are interpreting the reviewer's comment correctly, he/she feels that if the pMFC is shown to be a domain-general decision maker then our work on value-based decision making is automatically less interesting because of previous fMRI literature on the role of this area in perceptual decision making.

However, whether or not the brain networks and computational principles underlying value- and reward-based decisions mirror those reported in perceptual decision making is currently highly debated in the literature and is therefore an interesting question in its own right (i.e. it should not follow trivially from having provided evidence for one domain alone).

More fundamentally, however, the region of the pMFC we reported here differs both in location and functional role from what has previously been reported in the perceptual decision making literature. More specifically, work on perceptual decisions has implicated the pre-SMA in performing decision boundary adjustments within an accumulation-to-bound framework (Forstmann et al 2008, Ivanof et al. 2008, etc). In contrast our pMFC cluster encompasses the SMA proper and extends ventrally into the mid-cingulate cortex. Crucially, we provided compelling evidence that this region reflects the full temporal dynamics of the decision process (i.e. the moment-by-moment changes in the entire evidence accumulation time-series) that capture all relevant variability in the decision process (starting point, accumulation slope, boundary etc) rather than mere boundary adjustments (Supplementary Fig. 4e). This finding was enabled through an EEG-informed fMRI analysis approach that would not have been possible with stand-alone fMRI (see below). We view this as the main novelty of our manuscript over previous work (as also highlighted by the other two reviewers). We highlight this point further in the manuscript on pg. 10 and in the revised title of the paper, where we also removed the direct reference to the pMFC as it was not our intention to claim that this region is specific to value accumulation (following the reviewer's suggestion from the previous round of reviews).

Overall, the main focus of our work was not to offer evidence of a domain-general decision maker but to investigate the brain networks involved in evidence accumulation in value-based decisions. To accommodate the reviewer's earlier concerns we had made an effort to state explicitly that we chose to focus primarily on value- and reward-based decisions as it seems to us that efforts to identify potential accumulator regions in this domain received considerably less attention than in the long body of literature in perceptual decision making (see pg. 10, where we also discuss the role of pMFC across domains, including the previous work by Liu & Pleskac 2011 suggested by the reviewer). There are relatively fewer studies focusing on the mechanism underlying value-based as compared to perceptual decisions and indeed the relatively recent attempts to apply the drift diffusion paradigm to value-based

decision has not led to an to the identification of the locus of evidence accumulation for this kind of choices.

To do so here, we developed a novel methodology in which we combined computational modeling with simultaneously acquired EEG-fMRI to first identify an electrophysiologically-derived measure of the process of evidence accumulation in value-based decisions and subsequently exploit the trial-by-trial variability in this measure to drive the analysis of the fMRI data.

As decision-related activity is stochastic in nature and unfolds quickly in time, the temporal precision of the fMRI signal is not well suited for capturing its temporal dynamics. Instead, stand-alone fMRI often relies on average or individual trial estimates of decision and behaviourally relevant variables (e.g. drift rates and boundaries from model predictions and/or behavioural measures such as reaction times). This approach, however, inevitably conceals the endogenous variability in the decision process (i.e. within-trial variability) that might be required to explain BOLD activity from the relevant areas.

Here, by combining EEG with fMRI we are effectively considering both the “drift” and the “diffusion” part of the decision process by using the entire trace of the EEG activity in individual trials (which incorporates the moment-by-moment changes in the accumulation process) as a regressor of interest in the fMRI analysis. This in turn allowed us to identify the spatial locus of evidence accumulation in pMFC during value-based decision making. We now highlight the importance of our research methodology both in the new title of the paper and in the discussion section on pg. 8.

Future work will be required to show that the pMFC is a general-purpose decision maker, testing a larger set of tasks and effector modalities. However, collecting simultaneous EEG/fMRI on such a wide range of tasks and conditions is beyond the scope of this work.

Reviewer #3 (Remarks to the Author):

I would like to thank the authors for performing the additional analyses, and I am happy to see that the results remain, and if anything are even stronger.

We thank the reviewer for the positive assessment of our work and for the extremely constructive feedback in the previous round of reviews that have helped us further improve the manuscript.

Three relatively small issues remain in my opinion that could be addressed, related to my earlier points:

1. The authors fit additional models to test alternative hypotheses, but it seems they did not include the simpler model that only estimates drift, diffusion and NDT (ie neither leak nor threshold variability across participants). I think it would be good to exclude this model as well.

We thank the reviewer for suggesting this alternative model. We now ran this new model, which included only the three “core” parameters (drift, diffusion and NDT). We ran the new model on the same parameter intervals we used for k , σ , nDT , in our original model and found that it did not fit the data better than the original one (see figure below). When we formally compared the performance of the new model against the old ones using the Bayesian Information Criterion (BIC) the leaky model marginally outperformed all the others (summed BIC from leaky model: 696.53, BIC from ‘boundary’ model: 709.31, BIC from ‘joint’ model: 741.49, BIC from the ‘core’ model: 836.42).

Rebuttal Figure 1. Behavioral performance (blue circles) and modeling results (black and red crosses). Participants' average ($N = 21$) reaction time (RT) and accuracy (top and

bottom respectively) fitted by the “leak” model 1 and by the “core” model 2. The leak model clearly outperforms the “core” model.

2. How was the model comparison performed? Specifically, are the reported BIC values summed over participants?

The BIC was first computed for each participant’s data fit and then averaged across participants. The reported values were therefore average BIC over the population. We now however, report the summed BICs instead as it is somewhat more conventional. As expected results remained unchanged.

3. Supplementary figure 6a seems to suggest that the joint model fits actually worse than the leak model. Given that these are nested models, this should not be the case. The easiest solution to this issue would be to use the best fitting parameters of the leak model as initial estimates of the joint model, and show that individual estimates of the threshold parameter do not improve the fit.

We thank the reviewer for highlighting this issue. In fact the log likelihood term itself is higher for the “joint” than the “leak” model (-56.8 for the “joint” vs -101.4 for the “leak”), and it is only after we penalize for the extra parameter in the BIC computation that the “leak” model comes up on top. We interpret these results to mean that though the nested model fits the data marginally better the extra parameter does not improve the fit enough to justify selecting the less parsimonious model parameterization. We also realized that the grayscale color-coding of the models in the original supplementary figure 6a was slightly misleading. We therefore changed the color code to better appreciate the small differences between the two models (i.e. the fact that the joint model is overall slightly better). As the reviewer highlighted in the original review using the EEG signal itself as a regressor for the fMRI analysis ensures that potential misspecifications in the model have a lesser influence on the eventual inference. We now emphasize this more explicitly in the text on pg. 15.

Congratulations on this great work!

Signed,
Leendert van Maanen